# Cogito, Ergo Ludo: An Agent that Learns to Play by Reasoning and Planning

**Abstract:** The pursuit of artificial agents that can learn to master complex environments has led to remarkable successes, yet prevailing deep reinforcement learning methods often rely on immense experience, encoding their knowledge opaquely within neural network weights. We propose a different paradigm, one in which an agent learns to play by reasoning and planning. We introduce *Cogito, ergo ludo* (CEL), a novel agent architecture that leverages a Large Language Model (LLM) to build an explicit, language-based understanding of its environment's mechanics and its own strategy. Starting from a *tabula rasa* state with no prior knowledge (except action set), CEL operates on a cycle of interaction and reflection. After each episode, the agent analyzes its complete trajectory to perform two concurrent learning processes: *Rule Induction*, where it refines its explicit model of the environment's dynamics, and *Strategy and Playbook Summarization*, where it distills experiences into an actionable strategic playbook. We evaluate CEL on diverse grid-world tasks (i.e., Minesweeper, Frozen Lake, and Sokoban), and show that the CEL agent successfully learns to master these games by autonomously discovering their rules and developing effective policies from sparse rewards. Ablation studies confirm that the iterative process is critical for sustained learning. Our work demonstrates a path toward more general and interpretable agents that not only act effectively but also build a transparent and improving model of their world through explicit reasoning on raw experience.

## 1 Introduction

The quest to create intelligent agents (Sutton, 2022) capable of mastering complex, interactive environments has been a long-standing goal of artificial intelligence (Sutton & Barto, 2018). Landmark achievements, from Deep Blue's victory in chess to AlphaGo (Silver et al., 2016; 2017; 2018)'s dominance in Go, have demonstrated the power of computation and search (Sutton, 2019). More recently, large-scale deep reinforcement learning (RL) has produced agents with superhuman abilities in complex video games (Vinyals et al., 2019; Berner et al., 2019). These systems, however, often learn inefficiently through experience, requiring immense computational resources and encoding their strategic knowledge *implicitly* within the millions of *parameters* of a neural network, rendering their decision-making processes opaque.

The advent of Large Language Models (LLMs) presents a paradigm shift, offering a new foundation for agent design grounded in reasoning and explicit knowledge representation (DeepSeek, 2025; Gemini, 2025). While early LLM-based agents show promise (Gemini, 2025; Lin & Xu, 2025), they often lack a structured mechanism for continuous learning and adaptation. They may operate in a zero-shot capacity (Hu et al., 2025) or rely on simple memory retrieval (Gemini, 2025), but they do not fundamentally improve their internal model of the world's mechanics through experience. Similarly, while learned world models (Schrittwieser et al., 2020; Hafner et al., 2025; Richens et al., 2025) have enabled agents to plan in imagined futures, their models operate on uninterpretable latent states, shrouding their "understanding" of the world in a black box. This leaves a critical gap: the need for an agent that not only acts, but truly comprehends its environment in a way that is both effective and interpretable.

In this work, we introduce *Cogito, ergo ludo* (CEL), a novel agent architecture (Figure 1) that learns to master interactive environments not just by acting, but by *reasoning* and *planning*. We propose an agent that leverages an LLM to explicitly reason about its interactions, building and refining a human-readable "world model" (Ha & Schmidhuber, 2018; Sutton & Barto, 2018) of its environment and its own strategy from the ground up. Starting from a *tabula rasa* state with no prior knowledge of the game rules, CEL learns purely through a cycle of interaction and reflection, embodying the principle of learning by thinking.

**1. Conventional RL Paradigm**  **2. Zero-shot Reasoning Paradigm**  *3. "Cogito, ergo ludo" Paradigm*

Figure 1: A Comparison of Paradigms for Game-Playing Agents. This figure contrasts three distinct agent architectures: (1) the Conventional RL Paradigm that learns an implicit policy from rewards, by updating policy weights; (2) the Zero-shot Reasoning Paradigm that leverages a static LLM model for decision-making; (3) the *Cogito, ergo ludo* Paradigm, where the agent's policy is trained by RL while a persistent knowledge base (Rule & Playbook) is built and passed *across episodes.*

The cornerstone of CEL is its two-phase operational cycle. During an episode, the agent acts decisively by performing a lookahead search with *natural language*, using its current understanding of the world to predict the outcomes of its actions. Crucially, after each episode concludes, the agent enters a *Post-Episode Reflection* phase. In this phase, the LLM analyzes the trajectory of the preceding episode to perform two concurrent learning processes: *Rule Induction*, where it refines its explicit, language-based model of the environment's dynamics; and *Strategy and Playbook Summarization*, where it distills successful and unsuccessful patterns of behavior into an actionable strategic playbook. This refined knowledge base, both the rules of the world and the principles of how to act within it, directly informs the agent's decision-making in subsequent episodes.

We demonstrate the effectiveness of CEL across three distinct grid-world environments: the logical puzzle of Minesweeper, the navigation challenge of Frozen Lake, and the complex planning problem of Sokoban. Our experiments show that CEL successfully learns to master these tasks by autonomously discovering their rules and developing effective strategies. Ablation studies confirm that the iterative, reflective process of refining its internal knowledge is critical to its learning success. Furthermore, we provide qualitative evidence of the architecture's unique interpretability, showcasing the comprehensive, human-readable rulebooks and sophisticated strategic heuristics that it generates *entirely* from raw interaction. Our work presents a step towards agents that not only perform well, but also build a transparent and improving *understanding* of their world.

## 2 RELATED WORK

Our research builds upon decades of work in artificial intelligence, drawing from and extending three key areas: the paradigm of large-scale deep reinforcement learning, the development of learned world models for planning, and the nascent field of agents driven by Large Language Models.

**The Apex of Deep Reinforcement Learning.** Landmark achievements such as DeepMind's AlphaStar (Vinyals et al., 2019) and OpenAI Five (Berner et al., 2019) demonstrated that deep reinforcement learning (RL) could attain superhuman performance in complex real-time strategy games. These systems operate at a massive scale, training for thousands of GPU-years on billions of game frames. Their strategic acumen is implicitly encoded within the weights of enormous neural networks, learned through vast experience. While immensely powerful, this approach is characterized by high sample complexity and the opaque nature of the resulting policies. Our work diverges from this paradigm by pursuing a more sample-efficient and interpretable approach, where knowledge and strategy are explicitly represented in natural language. The AlphaZero algorithm (Silver et al., 2016; 2017; 2018) uses the power of Monte-Carlo Tree Search (MCTS) (Kocsis & Szepesvári, 2006) with a deep neural network, achieving superhuman performance in Chess, Shogi, and Go. But crucially, it was provided with a perfect model of the environment – the game rules, while our proposed architecture accumulates the knowledge of the game rules purely by interaction.

**Planning with Learned World Models.** MuZero (Schrittwieser et al., 2020) learned a latent model to predict future rewards, policies, and values, enabling effective lookahead search without being given the rules. This principle of learning and planning in imagined trajectories has been further advanced by algorithms like Dreamer (Hafner et al., 2025), which learns a robust world model that allows it to master a vast suite of diverse domains, from Atari to Minecraft, with a single set of

hyperparameters. Our Language-based World Model (LWM) shares this objective of predicting environmental dynamics. However, we draw a critical distinction: whereas the models in MuZero and Dreamer operate on uninterpretable latent states, our LWM is grounded in explicit, human-readable rules and transition dynamics that are themselves inferred from experience. This language symbolic foundation allows the agent to reason about and refine its understanding of the world's mechanics in natural language. These latent world-model agents are optimized for high-throughput interaction, whereas CEL targets symbolic text environments and emphasizes transparency and reusable language-level knowledge. We focus our comparisons on LLM-centric baselines in the same text-only interface and regard MuZero-style approaches as complementary rather than direct competitors.

**Language as the Algorithm vs. Language in the Architecture.** A distinct approach is Natural Language Reinforcement Learning (NLRL) (Feng et al., 2024), which seeks to fundamentally redefine the core components of RL, such as the value function and Bellman equation, entirely within the domain of natural language. In NLRL, the value of a state is not a scalar but a descriptive text, and the policy improvement step is performed by an LLM reasoning over these linguistic value judgments. While both approaches leverage LLMs for reasoning, our philosophy and architecture differ significantly. Rather than reformulating the RL algorithm itself into language, our framework treats the LLM as the orchestrator of a cognitive architecture composed of distinct, language-grounded modules.

**LLMs as Agent Architectures.** More recently, the advent of LLMs has catalyzed a new approach to agent design. Frameworks like GEM (Liu et al., 2025) and LMGame-Bench (Hu et al., 2025) provide environments and harnesses to evaluate LLM agents, highlighting challenges in perception, memory, and long-horizon planning. Gemini 2.5 Pro (Gemini, 2025) showcases its success in complete Pokémon game playing, demonstrating the strong zero-shot reasoning abilities of the frontier LLMs. A particularly relevant approach is PORTAL (Xu et al., 2025), which uses an LLM as a "policy architect" to generate behavior trees in a domain-specific language. Unlike PORTAL, our method uses LLM as the core for planning, acting and accumulating knowledge, where the LLM directly interacts with environments.

Our work builds upon this foundation but proposes a more comprehensive cognitive architecture. Our agent learns and maintains a suite of distinct, yet interconnected, cognitive components: an explicit world model of environmental dynamics, a set of game rules, a strategic playbook, and a language-based value function. The cornerstone of our method is the post-episode reflection phase, where the LLM analyzes interaction trajectories to iteratively and simultaneously refine both its understanding of the world's rules and its own strategic playbook. This creates a cycle of self-improvement that is explicit, interpretable, and broadly applicable to any interactive environment.

## 3 METHOD

We model the agent's interaction with its environment as a Markov Decision Process (MDP) (Puterman, 1990), formally defined by the tuple $(\mathcal{S}, \mathcal{A}, \mathcal{P}, \mathcal{R}, \gamma)$. In this framework, $\mathcal{S}$ represents the set of states, $\mathcal{A}$ the set of actions, and $\gamma \in [0, 1]$ the discount factor. The state transition function $\mathcal{P}(s_{t+1}|s_t, a_t)$ specifies the probability of transitioning to state $s_{t+1}$ from state $s_t$ upon taking action $a_t$. The reward function $\mathcal{R}(s_t, a_t)$ yields an immediate reward $r_{t+1}$. The agent's objective is to learn a policy $\pi(a|s)$, that maximizes the expected discounted return, defined as $G_t = \sum_{k=0}^{\infty} \gamma^k r_{t+k+1}$ (Sutton & Barto, 2018).

Our central methodological contribution is to employ a single Large Language Model (LLM), denoted $\mathcal{L}$, to instantiate and manage all of the agent's cognitive functions. Our framework moves beyond the static zero-shot paradigm by continuously training an LLM based on the outcomes of the agent's interactions, allowing it to improve its core reasoning and planning capabilities over time. We represent all information pertaining to the interaction: states ($s$), actions ($a$), rewards ($r$), inferred environmental dynamics ($\mathcal{G}$), and strategic guidelines ($\Pi$), as natural language strings. The agent's learning unfolds over a series of episodes, indexed by $k$, where each episode consists of discrete time steps, indexed by $t$. The LLM's reasoning process is made explicit through a chain-of-thought (Wei et al., 2022), which we denote by $C$. By reasoning and planning, the CEL agent $\mathcal{L}$ learns to interact with the environment and maximize its rewards.

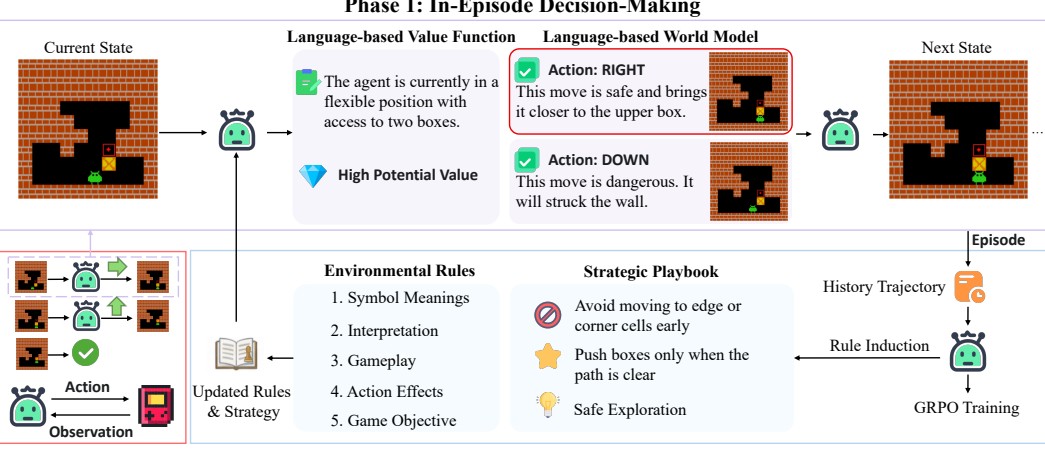

Figure 2: An overview of the *Cogito, ergo ludo* (CEL) agent's two-phase operational cycle. In Phase 1, the agent leverages its Language-based World Model (LWM) to predict the outcomes of potential actions and its Language-based Value Function (LVF) to evaluate the desirability of the resulting states, ultimately selecting the optimal action. In Phase 2, it reflects on the episode's trajectory to update its explicit knowledge base (Environmental Rules and Strategic Playbook). The agent continuously improves through this dual learning loop, which not only refines its explicit knowledge but also trains the LLM's internal parameters based on the final outcome.

## 3.1 LANGUAGE-BASED WORLD MODEL

The LLM functions as a world model (Ha & Schmidhuber, 2018; Hafner et al., 2025), tasked with predicting the dynamics of the environment. Given the current state $s_t$ and a candidate action $a_t$, the world model forecasts the subsequent state $\hat{s}_{t+1}$ and immediate reward $\hat{r}_{t+1}$. This prediction is conditioned on the agent state $s_t$, a potential action $a_t$, and the agent's current understanding of the environment's rules, $\mathcal{G}_k$. The model first generates a reasoning trace, $C_{WM}$ (for **W**orld **M**odel), before outputting its predictions:

$$(C_{WM}, \hat{s}_{t+1}, \hat{r}_{t+1}) \sim p_{\mathcal{L}}(\cdot|s_t, a_t, \mathcal{G}_k), \tag{1}$$

where $p_{\mathcal{L}}$ is the probability distribution over text sequences generated by the LLM. Critically, the outputs $\hat{s}_{t+1}$ and $\hat{r}_{t+1}$ are not structured data types (e.g., a state tensor or a scalar value) but are descriptive natural language strings, as illustrated in Figure 5.

This predictive capability is the foundation for explicit planning (Sutton & Barto, 2018). By querying the world model for each potential action, the agent can simulate and evaluate a set of possible future outcomes, a process that is central to its decision-making (Schrittwieser et al., 2020; Richens et al., 2025).

## 3.2 INDUCTION OF ENVIRONMENTAL DYNAMICS

Following each episode $k - 1$, the agent enters a reflective phase to refine its understanding of the environment's mechanics. The LLM performs rule induction by analyzing the trajectory of the concluded episode, $\tau_{k-1}$, in light of its previously held rules, $\mathcal{G}_{k-1}$. A trajectory is the sequence of state-action-reward tuples recorded during the episode:

$$\tau_{k-1} = \{(s_0, a_0, r_1), (s_1, a_1, r_2), \dots, (s_{T_{k-1}-1}, a_{T_{k-1}-1}, r_{T_{k-1}})\}. \tag{2}$$

The LLM processes this experiential data to generate an updated, more accurate set of rules $\mathcal{G}_k$:

$$(C_{\mathcal{G}}, \mathcal{G}_k) \sim p_{\mathcal{L}}(\cdot|\tau_{k-1}, \mathcal{G}_{k-1}), \tag{3}$$

where $C_{\mathcal{G}}$ is the reasoning trace for updating the environment's **G**overning dynamics (or **G**ame rules in game environments). We assume the agent begins with no explicit game knowledge (i.e.,

*tabula rasa*): despite the pretrained backbone, the agent's explicit game knowledge starts empty; in particular, the initial rule set $\mathcal{G}_0$ is empty, and all subsequent game knowledge is derived purely from interaction (Silver & Sutton, 2025).

### 3.3 Strategy and Playbook Summarization

In parallel with rule induction, the agent updates its high-level strategy. After episode $k-1$, the LLM synthesizes the trajectory $\tau_{k-1}$ and the final outcome $Z_{k-1}$ (e.g., success/failure, final score) to update a strategic playbook, $\Pi_k$. This process distills successful and unsuccessful patterns of interaction into explicit, actionable advice:

$$(C_\Pi, \Pi_k) \sim p_\mathcal{L}(\cdot|\tau_{k-1}, Z_{k-1}, \Pi_{k-1}), \tag{4}$$

where $C_\Pi$ is the reasoning trace for the **P**laybook update, and $\Pi_0$ is initialized with a general-purpose prompt.

This mechanism contrasts sharply with conventional reinforcement learning (Mnih et al., 2015; Espeholt et al., 2018), where strategy is implicitly encoded within the weights of a neural network. In such systems, adaptation is often slow, sample-inefficient (e.g., requiring millions of interaction frames (Mnih et al., 2015; Espeholt et al., 2018)), and opaque. Our approach externalizes strategy into an explicit, interpretable text playbook. Insights from a single episode can be immediately incorporated into the agent's prompt context for the subsequent episode. This mechanism facilitates rapid *in-context learning*, dramatically accelerating strategic adaptation.

### 3.4 Language-based Value Function

To guide its planning, the agent employs the LLM $\mathcal{L}$ as a language-based value function. This component estimates the value of a state $\hat{v}(s_t)$, by providing a qualitative, linguistic assessment of the long-term potential for success from that state. This evaluation is conditioned on both the current environmental rules $\mathcal{G}_k$ and the strategic playbook $\Pi_k$:

$$(C_V, \hat{v}(s_t)) \sim p_\mathcal{L}(\cdot|s_t, \mathcal{G}_k, \Pi_k). \tag{5}$$

Here, $C_V$ is the reasoning trace for the **V**alue estimation. This function provides the agent with a crucial heuristic by assessing the current state's long-term potential, which is essential for planning.

### 3.5 The Agent's Operational Cycle

The agent's operation is structured as a cyclical pipeline that alternates between two phases: in-episode decision-making and post-episode reflection (Figure 2). This architecture decouples rapid, step-by-step action selection from a more deliberate, offline knowledge consolidation process, with the LLM orchestrating both.

**Phase 1: In-Episode Decision-Making.** During an episode $k$, the agent operates with a fixed set of environmental rules $\mathcal{G}_k$ and a strategic playbook $\Pi_k$. At each time step $t$, it performs a structured reasoning process to select an action. First, the agent's *Language-based Value Function* (LVF) assesses the desirability of the current state, $s_t$, providing a high-level, holistic evaluation of its strategic potential. Concurrently, for each available action $a$, the agent's *Language-based World Model* (LWM) performs a one-step lookahead search to simulate the resulting state $\hat{s}_{t+1}$ and reward $\hat{r}_{t+1}$. The agent then commits to the action that the LWM predicts will lead to the most favorable outcome. The resulting $(s_t, a_t, r_{t+1})$ tuple is then recorded in the episode's trajectory $\tau_k$.

**Phase 2: Post-Episode Reflection and Refinement.** Once an episode concludes, the agent enters the reflection phase to update its internal knowledge base. It performs *Rule Induction* by providing the LLM with the complete trajectory $\tau_k$ and the prior rule set $\mathcal{G}_k$ to produce a refined set of rules $\mathcal{G}_{k+1}$. Concurrently, it engages in *Strategy and Playbook Summarization*, where the LLM processes $\tau_k$ and the final outcome $Z_k$ to distill key lessons, updating the playbook to $\Pi_{k+1}$. To reduce drift from individual anomalous episodes, we do not overwrite the rulebook with every newly induced rule set. Instead, we conservatively merge new rules into the existing rulebook, resolving conflicts and removing clearly inconsistent or redundant entries. The episode's final outcome (e.g., success or failure) provides a reward signal that is used to train the agent's core LLM, making it progressively more effective at planning and strategic reasoning in future episodes.

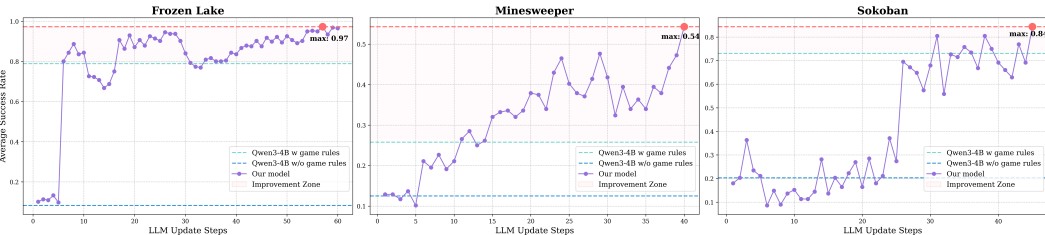

Figure 3: Learning curves of our agent on FrozenLake (left), Minesweeper (center), and Sokoban (right). The plots show the average success rate (y-axis) plotted against the number of LLM update steps (x-axis). Starting without any explicit rules, the agent's consistent improvement across these diverse tasks showcases the effectiveness of its autonomous rule discovery and policy learning. (see Figure 11 in Appendix D for curves with 95% confidence intervals).

This two-phase cycle enables the agent to both act decisively on its current understanding and systematically improve its environmental model and strategic acumen.

## 4 EXPERIMENTS

### 4.1 GAME ENVIRONMENTS

We evaluate our method on three classic grid-world environments: Minesweeper, Frozen Lake and Sokoban. These environments serve as common benchmarks for tasks with sparse rewards in reinforcement learning. In all experiments, the agent observes exactly the raw text grids emitted by the environments, with no additional handcrafted natural-language wrapper. Detailed descriptions of the environments are provided in Appendix B.

In our experimental setup, all three environments are configured with a sparse reward signal. The agent receives a reward only at the conclusion of the game, receiving +1 for successfully completing the objective and 0 otherwise. Crucially, we DO NOT provide the agent with any explicit game rules or the game names in the prompts; each environment is described only in generic, name-free terms. It must learn the dynamics of each environment solely through interaction, with its knowledge limited to the set of available actions. This setup, combining sparse rewards with unknown rules, presents a significant reasoning and planning challenge.

### 4.2 IMPLEMENTATION DETAILS

We conducted experiments using rLLM (Tan et al., 2025), backed by verl (Sheng et al., 2024). We use the Qwen3-4B-Instruct (Qwen, 2025) model to interact with the environments. We evaluate the performance over 32 randomly sampled seeds for each game. For each of these seeds, we conduct 8 independent trials, and report the average success rate over the total 256 playthroughs per game. The rule update frequency is set to once every 5 episodes. We use GRPO (Shao et al., 2024; DeepSeek, 2025) for LLM post-training and set the maximum response length to 8,192 tokens to encourage the model to think, reason, and plan. The outcome reward is used for optimizing the LLM.

### 4.3 RESULTS

Figure 3 illustrates our CEL agent's learning performance across the three environments. We compare CEL against two zero-shot baselines operating with and without the ground-truth game rules, respectively. Despite starting with no explicit game rules, the agent demonstrates a clear and positive learning trend in all tasks, validating the effectiveness of our interaction-reflection cycle. In the logical puzzle of Minesweeper, the agent exhibits steady improvement, with its success rate progressively climbing to a peak of 54%. Notably, this surpasses the 26% success rate of the baseline agent that was explicitly provided with the ground-truth game rules, suggesting that our method of autonomous rule discovery and strategy refinement leads to a more effective policy. A different learning dynamic emerged in the complex planning puzzle of Sokoban, where the agent's performance showed a distinct "breakthrough" pattern, increasing sharply to an 84% success rate after

an initial period of exploration. This highlights its ability to uncover critical insights for solving multi-step problems. The agent's efficiency was most apparent in the Frozen Lake navigation task, where it learned with remarkable speed to achieve a near-perfect success rate of 97% within the first 10 episodes. Collectively, these results showcase the general applicability and effectiveness of our approach, as it successfully masters diverse tasks ranging from logical deduction to long-horizon planning by autonomously discovering environmental rules and iteratively refining its own strategy from raw interaction.

## 4.4 ABLATION STUDY

We conducted an ablation study in the Minesweeper environment to test the necessity the component of the iterative Induction of Environmental Dynamics, with results shown in Figure 4. The baseline agent, operating without the Rule Induction mechanism ("w/o Rules"), exhibits a largely flat learning curve, with its success rate stagnating at a low level. This confirms that the ability to infer and utilize a model of the environment's dynamics is fundamental to achieving competence. A second variant, which performs Rule Induction only once and then uses a static rule set ("Rules induced once"), shows initial improvement but quickly stagnates and its performance degrades, suggesting its initial rules were incomplete or inaccurate. In stark contrast, our full CEL agent, which engages in the post-episode reflection and refinement phase to continuously update its rule set $\mathcal{G}_k$, shows a robust and sustained learning trajectory, significantly outperforming both

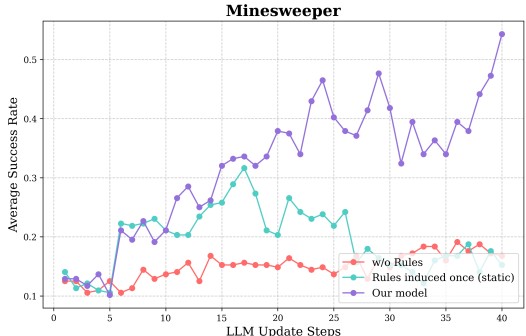

Figure 4: Ablation study illustrating the critical role of iterative **Induction of Environmental Dynamics**. The full model (blue) significantly outperforms variants with static rules (teal) or no rules (red), demonstrating that continuous refinement of the learned rulebook $\mathcal{G}_k$ is essential for sustained performance improvement.

ablated versions. This comparison unequivocally demonstrates that the iterative refinement of the agent's world model is a critical component of our architecture.

## 4.5 CASE STUDY

### 4.5.1 IN-EPISODE DECISION-MAKING

Figure 5 provides a qualitative snapshot of the agent's In-Episode Decision-Making process. The examples showcase how the agent performs a one-step lookahead search, a process relying on the synergy between its core cognitive components. First, the agent employs its Language-based Value Function (LVF) to produce a holistic, linguistic assessment of the current state's potential, $\hat{v}(s_t)$. In the Minesweeper example, it correctly identifies the state as having "high strategic value". Next, for each viable action, the agent utilizes its Language-based World Model (LWM), conditioned on its learned rules $\mathcal{G}_k$, to simulate the immediate future, predicting the next state $\hat{s}_{t+1}$ and reward $\hat{r}_{t+1}$. It accurately forecasts that the "(0, 3)" action in Minesweeper solves the puzzle. By comparing the predicted outcomes, the agent selects the action leading to the most favorable consequence, highlighting how its explicit, language-based reasoning drives intelligent planning.

### 4.5.2 AUTONOMOUS RULE DISCOVERY

Figure 6 presents an example of the agent's learned rulebook $\mathcal{G}_k$ for Minesweeper, a direct output of the Induction of Environmental Dynamics process. Synthesized from its interaction trajectory and starting from a *tabula rasa* rule, the generated rules are remarkably comprehensive and accurate, covering everything from *Symbol Meanings* to the ultimate *Game Objective*. As defined in our method, this explicit, human-readable rule set $\mathcal{G}_k$ is the critical information that grounds the agent's cognitive functions. It provides the foundation for the Language-based World Model to predict future states and for the Language-based Value Function to estimate state values, thereby enabling all subsequent planning.

| State | Prediction of the Value of Current States | Prediction of Immediate Consequences | |
|---|---|---|---|
| ```
    0 1 2 3 4
0         🚩
1     1 1 2 1
2
3   1 1
4     1
``` | The current state has a **high strategic value** because the only safe unrevealed cell is (0,3). Revealing it will complete the safe zones and leave only the mines unrevealed — which is acceptable in the game (as long as no mine is revealed). The game is effectively solved after (0,3) is revealed, as all other unrevealed cells are known to be mines and cannot be revealed without loss. | (0,3) | This cell is currently unrevealed and is safe (as deduced from constraints: adjacent to (1,2)=1 and (1,3)=2, and only one mine required in (0,2) and (0,3), but (0,2) is a mine, so (0,3) must be safe). Revealing (0,3) will result in a safe cell being revealed, which does not trigger a loss. This will complete the safe area and leave only known mines (at (0,2), (0,4), (4,0)) unrevealed. The game is now in a terminal state where all safe cells are revealed and no mine is exposed — thus, **a win**. |
| | | (0,2) | This cell is adjacent to (1,1)=1 and is the only unrevealed neighbor. Since (1,1)=1 has exactly one mine, and (0,2) is the only unrevealed adjacent cell, it must be a mine. Revealing it would expose a mine → immediate loss. |
| | | (0,4) | This cell is adjacent to (1,3)=2, which requires two mines. One mine is at (0,2), and (0,4) is the only other possible — so it must be a mine. Revealing it would result in a mine exposure → loss. |
| | | (4,0) | This cell is adjacent to (3,0)=1 and is the only unrevealed neighbor → must be a mine. Revealing it → loss. |

Figure 5: The agent's In-Episode Decision-Making process. At each step, the agent uses its Language-based Value Function (LVF) to assess the current state's value (middle column). It then employs its Language-based World Model (LWM) to predict the consequences of each action (right column). The agent selects the action leading to the outcome with the highest predict value.

| Rule Composition | Generated-Rule Definition (Minesweeper) |
|---|---|
| **1. Symbol Meanings** | – '.': Unrevealed cell (unknown content; may contain a mine or be safe).
– '0': Revealed safe cell (no adjacent mines).
– 'n' (where n > 0): Revealed cell with exactly 'n' adjacent mines (including diagonals).
– '*': Revealed cell that contains a mine (game over condition). |
| **2. Interpretation** | – The initial state is fully obscured (all cells are '.').
– After a valid action, the game reveals the state of the selected cell.
– If a revealed cell contains a mine ('*'), the game ends immediately — loss.
– If a revealed cell shows a number (e.g., 1, 2), it indicates exactly how many adjacent mines exist (including diagonals), but does not reveal their specific locations.
– A cell is "already revealed" if it has been previously opened and remains unchanged.
– Revealing a cell with value '0' provides no direct mine information but immediately reveals all adjacent unrevealed cells (via propagation). |
| ... | ... |
| **5. Game Objective** | – Win: All unrevealed cells are safely revealed (no mine is exposed), and the final board contains no '.' or '*'.
– Lose: A mine ('*') is revealed during a valid action.
– The game ends immediately upon revealing a mine.
– The game is "solved" when the player has fully revealed all safe cells without exposing any mines. |

Figure 6: An illustrative excerpt of the agent's learned rulebook $\mathcal{G}_k$ for Minesweeper, generated via the Induction of Environmental Dynamics process. Starting from no prior knowledge, the agent synthesizes a comprehensive and accurate set of rules from its interaction trajectory. Please refer to Figure 19 in the Appendix for the complete rulebook. All rules are generated by CEL. Blue highlighting marks a few representative cases.

### 4.5.3 EMERGENT STRATEGY AND PLAYBOOK GENERATION

In parallel with rule induction, our agent constructs a strategic playbook $\Pi_k$, via the Strategy and Playbook Summarization process, synthesized in Figure 7. As defined, the LLM analyzes an episode's trajectory $\tau_{k-1}$ and outcome $Z_{k-1}$ to distill experiences into actionable advice. The emergent knowledge exhibits a sophisticated hierarchy, from tactical *Methods* like Constraint Propagation to high-level *Principles* like Safe Exploration. The discovery of these expert-level heuristics from raw interaction highlights the agent's capacity for strategic abstraction. This explicit playbook $\Pi_k$ is then used alongside the rule set $\mathcal{G}_k$ in the next episode to condition the Language-based Value Function, enabling more nuanced, strategically-aware judgments and forming a direct feedback loop from experience to adaptation.

### 4.6 GENERALIZATION

To validate that CEL agent learns by understanding rather than memorization, we tested its generalization capabilities in two settings, summarized in Table 1.

| Derived Strategic Concept | | Evidence from Model (Minesweeper) |
|---|---|---|
| *Method* | Certainty | If a revealed number has exactly n adjacent unrevealed cells, those cells must contain the n mines. |
| | Elimination | Use numerical clues to deduce mine locations: When a number is revealed, use logical deduction to determine which unrevealed adjacent cells must contain mines. |
| | Constraint Propagation | Use pattern recognition and adjacent mine counts to deduce mine positions in complex configurations, especially when multiple revealed values provide overlapping constraints. |
| *Principle* | Maximum Information Gain | Avoid moving to edge or corner cells early — they have fewer neighbors and less information. |
| | Safe Exploration | Avoid blind exploration: Failed trajectories consistently involve random or peripheral that lead to mine exposure. Always validate safety before expanding. |

Certainty     Elimination     Constraint Propagation     Maximum Gain     Safe Exploration

Figure 7: A synthesis of the strategic playbook $\Pi_k$ for Minesweeper, generated via Strategy and Playbook Summarization. The agent distills both tactical *Methods* and high-level *Principles* from its gameplay experience. This explicit playbook is used to condition the agent's value judgments, enabling more strategically sophisticated decision-making. All rules are generated by CEL. Blue highlighting marks a few representative cases.

Table 1: The CEL agent's generalization performance across intra-game (unseen layouts) and inter-game (new game environments) settings. The results demonstrate the agent's strong generalization, showcasing both robust performance on unseen layouts (intra-game) and successful zero-shot transfer to novel environments (inter-game). Values in (.) show gain over the Zero-shot baseline, while those in [.] show change relative to the corresponding in-domain performance.

| Trained On | Tested On (Inter-Game) | | Intra-Game (Unseen layouts) |
|---|---|---|---|
| | Minesweeper | FrozenLake | |
| Zero-shot w/ Rule | 25.8 | 78.9 | - |
| Minesweeper | 53.5 (+27.7) | 97.3 (+18.4) | 50.4 [-3.1] |
| FrozenLake | 46.9 (+21.1) | 97.3 (+18.4) | 93.8 [-3.5] |

First, for **intra-game generalization**, we evaluated the agent on 32 new seeds that were entirely unseen layouts from those used during training. The agent maintained a high level of performance on these unseen 256 instances, confirming that it learns the game's fundamental principles rather than overfitting to the specific training levels. This highlights a fundamental paradigm difference from conventional reinforcement learning, which is notoriously hard to generalize to unseen domains. Instead of overfitting to learned patterns, the CEL agent's success on new layouts stems from its ability to apply an understanding of the game's rules to reason and plan effectively.

Furthermore, in the more challenging **inter-game generalization** setting, detailed in Figure 8, we tested a model trained on one game in the environment of another. Specifically, a Minesweeper-trained agent tested on Frozen Lake (left) and a FrozenLake-trained agent on Minesweeper (right) both show robust learning curves, despite their core model weights being frozen. This success indicates that the agent transfers not its knowledge of game-specific rules, but rather its fundamental ability to learn by reasoning and planning when faced and interact with a novel environment. The CEL agent thus demonstrates a sophisticated ability to generalize not the concrete dynamics of a game, but the abstract wisdom of how to *reason, plan* and then *act*.

## 4.7 BEYOND SINGLE-AGENT PUZZLES: A LONG-HORIZON GO-LIKE TWO-PLAYER GAME

Beyond the previous single-agent grid puzzles, we also consider a small Go-like board game to test whether CEL's reason–act–reflect loop can cope with two-player, long-horizon dynamics. The environment is a $9 \times 9$ two-player, perfect-information board game with a horizon of up to 100 moves. Two players alternately place stones on empty intersections, and the agent receives only textual board descriptions and a sparse terminal reward based on the final outcome. Unlike the single-agent puzzles, the next state now depends not only on CEL's move but also on the opponent's policy, so the transition dynamics are shaped by an explicit opponent rather than by a fixed environment.

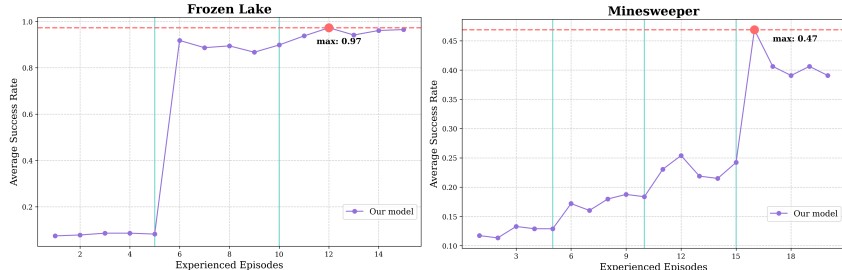

Figure 8: Inter-game generalization study showcasing adaptation to novel environments without model retraining. The plots show the agent's cross-game performance: a Minesweeper-trained agent on Frozen Lake (left) and a FrozenLake-trained agent on Minesweeper (right). In both evaluations, the core model weights remain frozen. The agent's adaptation relies solely on the iterative refinement of its explicit rulebook and strategic playbook every 5 episodes (indicated by cyan lines).

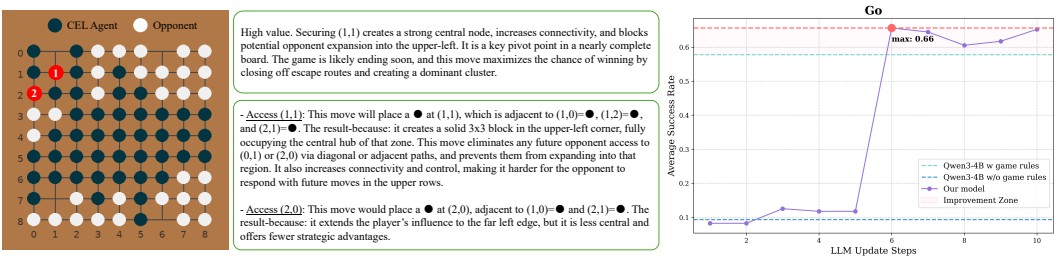

Figure 9: Go-like $9 \times 9$ adversarial board game. **Left:** Example board state and CEL's natural-language predictions of the value of the current state and the consequences of candidate moves, illustrating learned notions such as central influence, connectivity, and blocking the opponent's expansion. **Right:** Learning curves of CEL on the $9 \times 9$ Go-like game. CEL starts from a low win rate similar to Qwen3-4B without rules, then quickly improves and surpasses the rule-aware baseline, reaching a maximum win rate of 65.6%.

We use the same CEL architecture and training procedure as in other experiments and compare against baselines. As shown in Figure 9 (Right), the Qwen3-4B baseline w/o rules attains a low win rate of about 8%, while adding explicit rules raises performance to roughly 58%. Starting from a similar low win rate, CEL rapidly improves over training and reaches a maximum win rate of 65.6%, consistently outperforming both baselines against a rule-aware random-play opponent. The left panel of Figure 9 illustrates a typical late-game position and CEL's induced value estimate and move rationale, showing that the agent has learned to reason about central influence, connectivity, and future expansion opportunities. Although this setup uses a reduced board size and a simplified opponent, it demonstrates that CEL extends beyond single-agent puzzles to an adversarial, long-horizon board game while preserving interpretable, language-level rules and strategies.

## 5 CONCLUSIONS

In this work, we introduced *Cogito, ergo ludo* (CEL), a novel agent architecture that learns by explicitly reasoning about its environment. Through a unique two-phase cycle of in-episode planning and post-episode reflection, CEL autonomously constructs a human-readable world model and strategic playbook from raw interaction, starting from a tabula rasa state. Our results across several environments demonstrate that this "learning by thinking" approach allows the agent to master complex tasks while creating a transparent and auditable decision-making process. CEL marks a significant departure from opaque, brute-force learning paradigms. It validates language-based reasoning as a powerful foundation for building agents that are not only capable but also interpretable and trustworthy, opening compelling pathways toward hybrid systems that fuse CEL's explicit understanding with traditional architectural efficiency.

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

This appendix extends upon the main paper by providing additional details on our experimental setup, ablation studies, qualitative results, and implementation specifics.

- **Section A:** LLM usage policy for manuscript.
- **Section B:** Describes the game environments used in our experiments.
- **Section C:** Reports training details, hyperparameters, and the compute budget for all experiments.
- **Section D:** Details the evaluation protocol and the computation of confidence intervals for agent performance.
- **Section E:** Presents an additional supervised baseline based on behavioral cloning from expert demonstrations.
- **Section F:** Studies an action-only ablation trained with GRPO and explains why it fails to learn reliable policies.
- **Section G:** Controls game-specific prior knowledge via sanitized prompts and ablations to ensure the LLM is not simply recalling known games.
- **Section H:** Analyzes CEL's language-based value function and illustrates typical "value in words" feedback.
- **Section I:** Provides a brief failure analysis on the environments and highlights typical error modes of CEL.
- **Section J:** Discusses how CEL can be extended to broader classes of games and integrated with other agent modules.
- **Section K:** Shows results from an additional experiment on Minesweeper with an expanded training set, probing performance under more training seeds.
- **Section L:** Contains the full prompt templates used for in-episode decision-making, post-episode reflection, and rulebook merging.
- **Section M:** Provides qualitative examples of CEL's behavior, including decision-making traces, induced environmental rules, and strategic playbooks across different games.

## A  LLM USAGE

We utilized Large Language Models to assist in refining the language and improving the readability of this manuscript. The core intellectual contributions, including the conceptualization of ideas, experimental design, and analysis of results, were solely conducted by the authors.

## B  DETAILS OF ENVIRONMENTS

All game environments used in our experiments are from the TextArena (Guertler et al., 2025). Below are the descriptions for the specific environments and configurations used in this work. In all experiments, the agent's textual observations are exactly the raw ASCII grids provided by the environments. We do not introduce any additional handcrafted interpretation or rendering layer; CEL must infer the semantics of symbols purely from interaction.

**Minesweeper** is a logic puzzle where the objective is to clear a grid of all non-mine cells without detonating any mines. When a cell is revealed, it displays a number indicating how many adjacent cells contain mines, and the player must use this information to deduce the location of the mines. In our experiments, the game is configured on a 5×5 grid with 3 randomly placed mines.

**Frozen Lake** is a canonical grid navigation problem on a 6×6 grid where an agent must travel from a start tile to a goal tile, avoiding 6 randomly placed holes. In our deterministic setting, each action moves the agent exactly one cell in the chosen direction, removing the stochastic "slippery" nature often associated with this environment. This modification allows for a direct assessment of the agent's planning and rule-induction capabilities without the confounding factor of environmental randomness.

**Sokoban** is a classic puzzle game where the player must push all boxes to designated goal locations. The player can only push one box at a time and cannot pull boxes. This simple constraint creates a complex search space and necessitates careful, long-horizon planning to avoid irreversible states, such as trapping a box in a corner. The version used in our study is played on a 6×6 grid with a single box.

**Go** is a two-player board game in which players alternately place stones and the final result depends on captured stones and controlled territory. In our experiments, we use a simplified 9×9 Go-like environment with a horizon of 100 moves, where CEL plays against a rule-aware random-play opponent that samples uniformly from legal moves. States and actions are represented in text, and the agent receives only a sparse terminal reward based on the final game outcome (win or loss).

## C  TRAINING DETAILS, HYPERPARAMETERS, AND COMPUTE BUDGET

Our implementation is based on the open-source rLLM framework for post-training language agents with reinforcement learning. This section details the CEL training loop, the main hyperparameters used in all experiments, and the overall compute budget.

### C.1  CEL TRAINING LOOP

The full training loop of CEL is summarized in Figure 10. Each iteration proceeds as follows:

1. **Environment interaction.** The current policy interacts with the environment to collect trajectories consisting of observations, actions, and rewards.

2. **Language-based world modeling and rollouts.** Conditioned on the current rulebook and strategic playbook, the backbone LLM, acting as a Language World Model and Language Value Function, generates natural-language reasoning tokens, candidate actions, and value estimates.

3. **Reflection and knowledge update.** A reflection step analyzes successful and failed trajectories and updates the rulebook and strategic playbook accordingly, refining both the induced game rules and high-level strategies.

4. **GRPO update.** A GRPO trainer updates the backbone LLM parameters using the collected reasoning traces, actions, and rewards. The updated policy is then used in the next interaction phase.

This loop is repeated for all games with the same architecture and training procedure.

### C.2  HYPERPARAMETERS AND COMPUTE BUDGET

Table 2 lists the main hyperparameters used in our experiments. Unless otherwise specified, the same settings are used across all environments. All other low-level system and optimization settings (e.g., gradient checkpointing, FSDP parameter/optimizer offloading, asynchronous rollout engine) follow the default configuration of the rLLM framework and are kept fixed across experiments.

Training a single game configuration (e.g., Minesweeper) requires approximately 20 wall-clock hours on 8 GPUs (about 160 GPU-hours). The training cost scales approximately linearly with the number of sampled environment steps. All experiments are run on a small cluster of commodity GPUs and do not rely on any specialized hardware.

## D  CONFIDENCE INTERVALS FOR AGENT PERFORMANCE

For completeness, we report the evaluation protocol and confidence-interval computation used in the learning curves with uncertainty estimates (Figure 11).

**Evaluation protocol.** After every GRPO update of the CEL agent, we evaluate the current checkpoint in each environment on a fixed set of $N = 32$ level seeds, each corresponding to a different randomly generated level layout of the same game. For each seed $i$, we run 8 independent rollouts

Figure 10: Overview of the CEL training loop. During rollout, the CEL agent interacts with the game environment, receiving only textual states and scalar rewards, and uses a Language World Model and Language Value Function, conditioned on its induced game rules and strategic playbook, to generate natural-language reasoning tokens and actions. Multiple trajectories starting from the same initial state are sampled and their rewards are aggregated into advantages, which are fed to a GRPO trainer to update the backbone LLM weights. In parallel, a rule-induction module uses the sampled trajectories to refine the learned rulebook and strategic playbook. No ground-truth rules or demonstrations are provided by the environment.

| Category | Hyperparameter | Value |
|---|---|---|
| Model / Algorithm | Backbone model | Qwen3-4B-Instruct-2507 |
| | RL algorithm | GRPO |
| | Advantage estimator | GRPO (step-wise, MC return) |
| Optimization | Learning rate | $1 \times 10^{-6}$ |
| | Train batch size | 32 |
| | Max gradient norm | 10 |
| | KL coefficient | 0.001 |
| Rollout / Sampling | Temperature (train / val) | 0.6 |
| | Top-$p$ (train / val) | 0.95 |
| | Samples per prompt ($n$) | 8 |
| Length / Context | Max prompt length | 8,192 tokens |
| | Max response length | 8,192 tokens |
| Training schedule | Seeds $\times$ episodes per seed | $32 \times 8$ |
| Environment / Agent | Max env/agent steps per episode | 40 (Minesweeper) |
| | Trajectory timeout | 1,200 s |

Table 2: Main hyperparameters used for training CEL. Unless otherwise noted, these values are shared across all environments.

under the same policy parameters but different environment randomness and compute the empirical success rate $s_i \in [0, 1]$ for that seed. The scalar performance statistic at a given update step is then the average success rate across the 32 seeds:

$$\hat{m} = \frac{1}{N} \sum_{i=1}^{N} s_i, \quad N = 32.$$

**Confidence intervals.** We approximate a $95\%$ confidence interval over seeds using the standard normal approximation,

$$\hat{m} \pm 1.96 \cdot \frac{\hat{\sigma}}{\sqrt{N}},$$

where $\hat{\sigma}$ is the sample standard deviation of $\{s_i\}_{i=1}^{N}$. In this construction, multiple rollouts on the same level seed are first aggregated into a single success rate $s_i$, and the confidence interval captures variability due to both learning stochasticity and differences in seed difficulty. The same protocol is applied to the strongest baseline (Qwen3-4B with game rules).

In Figure 11, the solid purple curve denotes the mean performance of CEL and the purple shaded band shows its $95\%$ confidence interval. The dashed green curve and light-green band denote the mean and $95\%$ confidence interval of the Qwen3-4B with game rules baseline, and the dashed blue curve denotes Qwen3-4B without rules. Across all three environments, the CEL curve eventually rises above the Qwen3-4B with game rules confidence band and remains there at convergence. In particular, in Minesweeper and Sokoban the final CEL interval lies well above the baseline interval, and in Frozen Lake the CEL agent further improves upon an already strong baseline and maintains a higher success rate thereafter. These results indicate that the gains of CEL over Qwen3-4B with game rules are consistent across seeds and are not an artifact of high variance or a small number of especially favorable seeds.

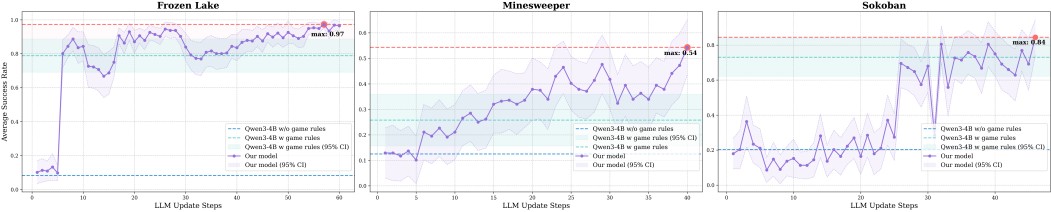

Figure 11: Average success rate of CEL and Qwen3-4B baselines in the three environments. The solid purple curve and band show CEL with $95\%$ confidence intervals; dashed lines are Qwen3-4B without game rules (blue) and with game rules (green, with $95\%$ confidence-interval band). Across all three environments, CEL consistently improves with training and eventually surpasses the Qwen3-4B with game rules baseline, with the final confidence intervals indicating a clear advantage for CEL.

# E    BEHAVIORAL CLONING BASELINE

To assess how CEL compares to supervised imitation learning, we consider a behavioral cloning (BC) baseline on the Minesweeper environment. The BC model uses the same backbone as CEL (Qwen3-4B-Instruct-2507) and is fine-tuned on a set of 2,000 expert state–action pairs generated by a scripted expert derived from a stronger model (Qwen3-30B).

We train the BC model with a standard cross-entropy loss, using the same input representation and tokenizer as CEL. The model fits the training data with high action-prediction accuracy, indicating that it can memorize the provided demonstrations. However, when evaluated on the same held-out test distribution as CEL (procedurally generated levels of the same game), the BC policy achieves essentially zero success rate, failing to generalize even to new instances of the same game.

In contrast, CEL is trained only from sparse binary success/failure feedback in the same Minesweeper setting, without access to expert demonstrations. Through its interaction–reflection loop, CEL induces an explicit rulebook and strategic playbook and achieves strong success rates on held-out boards. This comparison suggests that, in our Minesweeper setting, directly imitating a small set of expert actions is insufficient, while CEL trained from sparse feedback generalizes well to new boards.

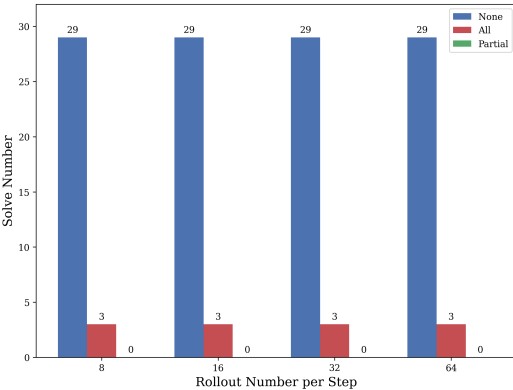

Figure 12: Distribution of rollout outcomes for the Action-only model. Across all group sampling sizes (8 to 64), outcomes are polarized into "All" (all succeed) or "None" (all fail). The number of "Partial" rollouts, which are required for GRPO to learn, is consistently zero, causing training failure.

## F    LIMITS OF AN ACTION-ONLY AGENT TRAINED WITH GRPO

To understand the necessity of cognitive components and chain-of-thought reasoning, we ablated it with a simplified "Action-only" agent that directly outputs actions. When attempting to train this agent with GRPO, we observed a consistent training failure. The reason lies in the extreme polarization of rollout outcomes, as shown in Figure 12. GRPO requires batches with mixed results (i.e., "partially successful") to derive a learning signal. However, across all tested sampling sizes (8 to 64), the outcomes for the Action-only agent were always binary: for any given seed, all rollouts in a batch either succeeded or failed. As a result, the number of partially successful rollouts, i.e., the sole source of a viable training signal, was consistently zero. This lack of comparative data within batches starves the GRPO algorithm of a gradient, leading to a breakdown in training and highlighting the critical role of nuanced reasoning traces in enabling effective optimization.

## G    CONTROLLING GAME-SPECIFIC PRIOR KNOWLEDGE

To address concerns that the pretrained backbone might simply recall prior knowledge of Minesweeper, FrozenLake, or Sokoban, we carefully control what information is made available in the prompts and provide empirical checks.

**Sanitized, name-free prompts.** At the beginning of training, the agent observes only the raw text-based grid of the current state and generic instructions about interacting with an unknown environment. In particular, the prompts do not contain the canonical game names "Minesweeper", "Frozen Lake", or "Sokoban".

**Baseline under sanitized prompts.** The zero-shot baseline Qwen3-4B w/o game rules uses exactly the same sanitized, name-free prompts as CEL but does not maintain any explicit rulebook or play-book. Under this setting it achieves very low success rates across all three environments (Figure 3), indicating that the pretrained model cannot solve these tasks by simple recall when given only the sanitized prompts.

**Effect of naming the game.** To probe how much information the name alone carries, we add a baseline Qwen3-4B w/o rules + game name, which is identical to Qwen3-4B w/o game rules except for a single sentence that names the environment (for example, "You are playing Minesweeper."). In this condition, zero-shot success rises to 10.94% on Minesweeper, 38.28% on FrozenLake, and 48.05% on Sokoban. This contrast shows that the canonical names encode strong priors in the pretrained model and confirms that name-free prompts are needed to fairly evaluate CEL's ability to induce rules and strategies from scratch.

# H  VALUE IN WORDS: LANGUAGE-BASED FEEDBACK FROM CEL

The language-based value function (LVF) is a component that, given the current state together with the induced rules and playbook, produces a qualitative assessment of the state's long-term potential in natural language. In our implementation, each LVF output is a short free-form text segment that begins with a coarse label, followed by a brief textual explanation of why the state is promising or risky. We treat this entire description (the coarse label together with its explanation in natural language) as the LVF's value expression.

Figure 13 shows several representative LVF outputs together with a brief description of which aspects of the state they emphasize. The examples show that the LVF naturally refers to multiple interacting aspects of a state (for instance, progress toward the goal, safety and future options, or longer-horizon opportunities), indicating that language-valued evaluations can capture richer structure than a small, fixed set of discrete value labels.

| Emergent Value Pattern | Example LVF output (short excerpt) | Our analysis: qualitative value dimensions |
|---|---|---|
| 1. Goal proximity & reachability | *"High value. The current state offers a clear path toward G... this proximity provides immediate strategic value and strong potential for further progress."* | The LVF here expresses that state value depends on distance to the goal and the existence and quality of a feasible path, not just the immediate reward. States near a solvable goal configuration are treated as intrinsically high-value. |
| 2. Safety assessment: risk vs. robustness | *"Medium value. Moving right increases exposure to the block, while moving down keeps a safer and more flexible position for future exploration."* | The LVF here expresses a comparison of risk and future robustness between alternative actions. Value is higher for states that maintain safety and preserve many future options, and lower for states that move closer to hazards. |
| 3. Progress vs. dead end | *"This action does not help push a block and moves the player away from it. It brings no strategic benefit and may lead to a dead end."* | The LVF here expresses that states are valuable only if they enable meaningful progress toward the objective and avoid irreversible dead ends. Reachable but non-productive positions are explicitly treated as low-value. |
| 4. Long-horizon planning & chain value | *"Medium to low value. The block cannot be pushed directly into the goal, but a viable win sequence still exists if the player first moves behind it and then pushes it down."* | The LVF here expresses that value depends on the existence of multi-step winning plans. Even with zero immediate reward, a state remains valuable if it still supports a plausible path to victory, reflecting look-ahead over future trajectories. |
| 5. Uncertainty & reachability under ambiguous rules | *"Value: low to medium. The player can push the adjacent block, which might open a path if the goal lies beyond it; otherwise the position is effectively stuck."* | The LVF here expresses epistemic uncertainty: value is discounted because goal reachability is unclear. The same move may either improve or ruin the position, so the state is assigned an intermediate value reflecting this uncertainty. |
| 6. Positional control of key regions | *"High value. Controlling the central column is critical; from here the player can reach both left and right blocks, giving a strong positional advantage."* | The LVF here expresses that value can come from control of strategically important areas, not just immediate reward. Central positions that connect multiple future opportunities are treated as high-value due to their durable positional advantage. |
| 7. High-value targets & timing | *"High value. The G and P tiles form high-value targets, and the current state offers a good opportunity to move toward them before the path becomes blocked."* | The LVF here expresses that value depends on both what high-value targets exist and when they can be reached. States are valuable when they provide a timely opportunity to approach important tiles that may later become unreachable (an opportunity window). |

Figure 13: Examples of language-valued feedback produced by CEL. Each row shows a representative snippet of the LVF output (middle column) together with a brief description of the qualitative value aspects it emphasizes (right column). These value expressions distinguish fine-grained differences in state quality and capture nuanced value structure, providing a richer signal than a small discrete set of value labels.

# I  FAILURE ANALYSIS OF HARD CASES

We conducted a brief failure analysis by manually inspecting failed episodes near the end of training and observed several recurring error modes. These cases help clarify where CEL currently struggles despite having induced reasonable local rules.

First, in FrozenLake the agent can over-generalize from a few sparse negative experiences and induce rules that are too coarse. For example, we sometimes see playbook entries that effectively discourage visiting an entire region of the grid after a few unlucky falls into holes in that area, even though some layouts require crossing that region to reach the goal. In such cases the induced strategy becomes overly conservative: the agent prefers to remain in safe but unproductive parts of the map and fails to commit to the unique risky path that is actually needed to succeed.

Second, in Sokoban the hardest failures arise in layouts with long, irreversible action chains. A move that clearly improves the local situation, such as pushing a box closer to a goal tile or clearing a corridor, can still commit the agent to a configuration that only becomes a deadlock many steps later. Once a box is pushed into a corner or along a narrow hallway, the position may be technically unsolvable although this is not immediately apparent from the local view. In many of these cases the relevant local rules (how boxes move, when a cell is blocked, what makes a box irrecoverable) are already present in the rulebook and reflected in the LVF, but the decision procedure still prefers locally attractive actions that subtly reduce the number of viable solution branches.

We also observe milder versions of these phenomena in Minesweeper. Occasionally CEL learns rules that slightly overestimate the risk of uncovering certain border cells and therefore chooses low-information moves that avoid progress, especially on larger boards where many patterns are rare. Here the failure is not that the rules are wrong in a single state, but that they are applied too rigidly and prevent the agent from gathering the evidence needed to refine them.

Taken together, these patterns highlight two main challenges for the current instantiation of CEL: the induced playbook can be sensitive to rare but misleading experiences, and action selection can struggle with rare long-horizon traps in environments with irreversible dynamics. We view these as opportunities for future work, for example by using stronger backbone LLMs and integrating deeper or more selective lookahead mechanisms in domains where long-term consequences are critical.

## J EXTENDING CEL TO BROADER CLASSES OF GAMES

While our main experiments evaluate CEL in controlled single-agent grid environments with symbolic state and sparse rewards, the same architecture naturally extends to broader classes of games and composes cleanly with other agent modules. In particular, CEL scales most directly to discrete, turn-based settings with symbolic or symbolizable state representations, and can serve as a high-level reasoning module that provides rules, value estimates, and strategic guidance on top of perception, search, control, or belief-tracking components in richer environments.

For larger symbolic puzzles and grid-based MDPs (e.g., maze and navigation domains, 2048-style tile games, grid-based resource-collection or delivery tasks), the state can be expressed as structured textual or grid descriptions and actions remain discrete. In these settings the LWM, LVF, and rulebook/playbook can be reused without architectural changes, and our size- scaling and horizon-scaling experiments on Minesweeper and Frozen Lake already indicate that CEL can cope with larger boards and longer episodes in this family. Moving to more complex grids primarily requires richer symbolic encodings (for example, for local patterns or objects) and adjusted horizons, while keeping the reason–act–reflect loop intact.

For combinatorial, perfect-information board games (e.g., Go and chess), CEL is well-suited as a high-level inducer of rules and strategies operating on symbolic board descriptions. Our Go-like two-player experiment in Section 4.7 shows that CEL can learn non-trivial strategies and outperform Qwen baselines against a rule-aware random opponent in a long-horizon setting. Scaling toward richer board games (e.g., larger boards or stronger opponents) would mainly require more structured symbolic encodings of game state (such as groups, liberties, or tactical features) and more expressive backbone LLMs to support deeper, more precise reasoning; the CEL loop itself remains unchanged.

In multi-agent or partially stochastic symbolic games (e.g., cooperative grid-worlds, capture-the-flag arenas, or turn-based card games with shared public information), CEL can be extended by conditioning the LWM and LVF on agent identities, roles, and shared state, and by treating the rulebook and playbook as explicit multi-agent knowledge (conventions, coordination patterns, role-specific guidelines). Partial observability and exogenous stochasticity can be handled by composing CEL with a separate belief-tracking module. For example, in a card game or a strategy game with fog-of-war, a dedicated module can maintain a latent belief over hidden cards or unseen units and periodically summarize it into a short textual description (e.g., which regions are likely unsafe or which cards the opponent is likely to hold), which is then fed to CEL as additional context.

Finally, in high-bandwidth, continuous-control, visually rich environments (e.g., FPS-style games), we do not expect CEL in its current text-only form to directly output low-level motor commands. Instead, we view CEL as a high-level reasoning and memory component on top of perception, search, and control stacks. A perception module would convert raw observations into compact symbolic descriptions; a learned motor policy would handle fine-grained continuous control; and CEL's LWM, LVF, and rulebook/playbook would operate on this abstract state space to induce rules, evaluate high-level options, and output goals, subgoals, or tactical plans. Tree search or other planners can then use CEL's world model and value function as language-level heuristics and priors, while explicit rules and strategies provide interpretable guidance and long-term memory for the overall agent.

## K PERFORMANCE WITH AN EXPANDED TRAINING SET

To further evaluate the scalability and generalization capabilities of our CEL agent, we conducted an additional experiment on the Minesweeper environment. We expanded the set of training layouts by increasing the number of unique seeds from 32 (used in the main experiments) to 128.

The results are presented in Figure 14. The agent demonstrates a notable improvement in performance, with its peak success rate climbing from 54% (as reported in Figure 3) to a new maximum of 62%. This finding suggests that exposure to a more diverse set of game scenarios directly enhances the agent's core reasoning and planning capabilities. This confirms that the agent is developing a robust, generalizable problem-solving model for the game, rather than overfitting to a limited number of specific layouts.

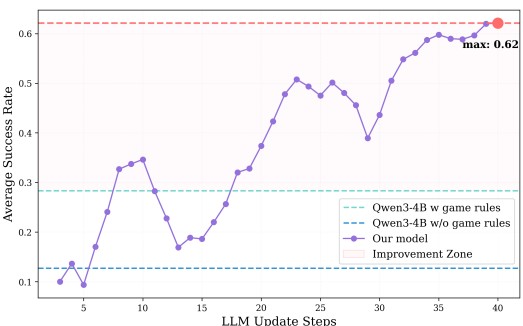

Figure 14: Learning curve for the CEL agent on Minesweeper when trained on an expanded set of 128 unique seeds. The agent achieves a new peak success rate of 62%, surpassing the performance observed with 32 seeds.

## L PROMPT TEMPLATES

In this section, we present the core prompt templates used by the CEL agent. Figure 15 shows the template for in-episode decision-making, Figure 16 shows the template for post-episode reflection, and Figure 17 shows the template for conservatively merging newly induced rules into the persistent rulebook.

## M ADDITIONAL RESULTS

To illustrate the explicit and interpretable knowledge base generated by our CEL agent, we provide concrete examples of Decision-Making processes (Figure 18), learned environmental rules (Figure 19, Figure 20, Figure 21) and strategic playbooks (Figure 22, Figure 23) for the Minesweeper, FrozenLake and Sokoban environment.

---

### LANGUAGE-BASED WORLD MODEL THINKING PROMPT - SYSTEM

---

You are an agent playing a game on a grid, acting as a reasoning engine.

**Game Context:**
Your decisions are based on two key pieces of information:
**These rules and strategies may be incomplete or incorrect.**
- **Current Game Rules:** Your best guess of how the game works.
{game_rules}
- **Strategic Playbook:** Learned strategies and principles for achieving success.
{strategic}

**Valid Actions:**
Your only way to interact is to **access an element** on the grid. You must specify its coordinates in the format `<answer>(row, col)</answer>`.

**Instructions:**
1. **Analyze State:** Summarize the current state.
2. **Predict Long-term Value of Outcomes (Value Function Evaluation):** Evaluate the strategic value and potential of the current state for the future.
3. **Predict Immediate Consequences (World Model Simulation):** For top 2 candidate actions, predict their consequences using a "result-because" structure.
4. **Select the Best Action:** Based on the predicted consequences, choose the action that leads to the most advantageous future state.

Your response must strictly follow the format below:

<reason>
**1. Analysis of the Current State:**
[Summary of the board state.]

**2. Prediction of the Value of Current States:**
[Provide an assessment of the current state's strategic value.]
- **Value:** High value. Securing guaranteed points creates a dominant position for winning.

**3. Prediction of Immediate Consequences:**
[Analyze ONLY the top 2 candidate actions using the "result-because" structure.]
- **Access (row_A, col_A):** ...
- **Access (row_B, col_B):** ...
</reason>
<answer>(row, col)</answer>

---

### LANGUAGE-BASED WORLD MODEL THINKING PROMPT - USER

---

Turn {turn}:

Observation is:
{current_observation}

---

Figure 15: The prompt template for in-episode decision-making (Phase 1). It instructs the LLM to evaluate the current state, assess their strategic value (LVF) and predict action outcomes (LWM).

1134
1135
1136
1137
1138
1139
1140
1141
1142
1143
1144
1145
1146
1147
1148
1149
1150
1151
1152
1153
1154
1155
1156
1157
1158
1159
1160
1161
1162
1163
1164
1165
1166
1167
1168
1169
1170
1171
1172
1173
1174
1175
1176
1177
1178
1179
1180
1181
1182
1183
1184
1185
1186
1187

---

### RULE AND PLAYBOOK SUMMARIZATION PROMPT - SYSTEM

You are a chief scientific strategist and master tactician. Your mission is to analyze extensive field data from numerous operations to distill and refine the **Master Rulebook** of a complex game.

You will be presented with a large collection of **highly successful trajectories** and **critical failure trajectories**, collected over a long period.

Your **primary task** is to perform a deep, comparative analysis to understand the fundamental principles of victory and defeat. You must act as a grand strategist, looking for universal patterns and high-level causal relationships. Your goal is to synthesize these insights to produce the **next generation's Master Rulebook**, making it more robust, accurate, and effective.

[Core Principles]
- **Think Long-Term:** Focus on universal, strategic truths that are consistently validated across many diverse scenarios. Ignore circumstantial flukes.
- **Learn from Contrast:** The most critical insights come from identifying the key strategic differences that separate winners from losers.
- **Synthesize and Consolidate:** Your output must be a single, unified, and improved Master Rulebook. Do not simply copy rules; forge a more perfect theory from all available evidence.
- **Be Authoritative and Concise:** Your rules should be stated as clear, definitive principles.

**Your output MUST be a single, consolidated `<rule>` block representing the new Master Rulebook.**

```
<rule>
<game_rules>
**1. Symbol Meanings:** [Define the unchanging, intrinsic properties of game elements.]
**2. Information & Interpretation:** [Define how elements reliably inform about the game state.]
**3. Gameplay & Actions:** [Define the core mechanics and interactions.]
**4. Action Effects:** [Describe the predictable outcomes of actions.]
**5. Game Objective & Termination:** [State the ultimate win/loss conditions.]
</game_rules>
<strategic>
**1. Core Strategies:** [Describe foundational, high-level strategic priorities that lead to victory.]
**2. Tactical Tips:** [List widely applicable, advantageous situational plays.]
</strategic>
</rule>
```

### RULE AND PLAYBOOK SUMMARIZATION PROMPT - USER

**[Task]: Analyze the following successful and failed gameplays to refine the Master Rulebook.**

### Evidence File 1: Successful Trajectories (Score: 1)
{positive_trajectories_data}

### Evidence File 2: Failed Trajectories (Score: 0)
{negative_trajectories_data}

**[Instruction]: Now, based on your comparative analysis of the evidence, provide the single, updated Master Rulebook.**

Figure 16: The prompt template for the Rule Induction and Playbook Summarization process (Phase 2). It guides the LLM to analyze a completed episode's trajectory and refine its explicit model of the environment's dynamics.

---

### RULE AND PLAYBOOK MERGE PROMPT - SYSTEM

You are an AI assistant functioning as a Rulebook Fusion Module.
Your sole purpose is to intelligently merge two versions of a game's rulebook: an established 'Base Version' and a new 'Proposed Version'.

Your primary directive is to produce a refined 'Final Version' that prioritizes stability and logical consistency. You must operate under the assumption that the 'Base Version' is fundamentally sound and effective, while the 'Proposed Version' contains potentially valuable but unverified updates.

[Operational Principles]

1. **Prioritize the Base Version:** The 'Base Version' is your foundation. Your output must be an evolution of it, not a replacement. Preserve its structure and core principles unless there is a compelling reason to change.

2. **Be Conservative:** Adopt changes from the 'Proposed Version' with high scrutiny. Only integrate modifications or additions that represent clear, unambiguous, and essential improvements. If a proposed change is radical, confusing, or contradicts the core logic of the Base Version, you must discard it and default to the Base Version's original rule.

3. **Ensure Coherence:** The 'Final Version' must be a single, coherent, and logical document. After merging, ensure there are no redundant, conflicting, or nonsensical rules.

**Your output MUST be a single, complete, and consolidated `<rule>` block representing the Final Version.**

---

### RULE AND PLAYBOOK MERGE PROMPT - USER

**[Task]: Merge the following two rulebook versions into a single, updated Final Version.**

### Base Version (The stable, foundational rulebook)
{base_rulebook}

### Proposed Version (The new draft with potential updates)
{proposed_rulebook}

**[Instruction]: Now, provide the single, merged, and updated Final Version in the required format.**

---

Figure 17: The prompt template for the rulebook and playbook merge step. It instructs the LLM to reconcile newly proposed rules and strategies with the existing rulebook, resolve contradictions, remove redundancy, and produce a refined, stable set of explicit game knowledge.

| State | Prediction of the Value of Current States | | Prediction of Immediate Consequences |
|---|---|---|---|
|  | **High value**. The player is in a direct line of action to push a box into the goal. The box is one step away from the goal, and the direction of push is clear. This is a high-probability win scenario with only one valid move required to succeed. The game is on the verge of being solved, and no further steps are needed if the push is valid. | **RIGHT** | The box at (4,3) will be pushed to (4,4), which is the goal ('G'). Since the cell behind the box (in the direction of movement) is empty (specifically, a goal cell), this move is valid and will mark the box as '√'. The game will immediately transition to a **"Game solved"** state. This move achieves the win condition in one step. |
| | | UP | The player would move to (3,2), which is currently empty ('.'). This does not advance the box or get closer to a goal. It reduces the player's proximity to the box and creates a non-productive move. This move leads to a loss of progress and wastes a step, especially since a win is immediately available via RIGHT. |

Figure 18: An example of a Decision-Making process for Sokoban environment.

| Rule Composition | Generated-Rule Definition (Minesweeper) |
|---|---|
| 1. Symbol Meanings | – '.': Unrevealed cell (unknown content; may contain a mine or be safe).
– '0': Revealed safe cell (no adjacent mines).
– 'n' (where n > 0): Revealed cell with exactly 'n' adjacent mines (including diagonals).
– '*': Revealed cell that contains a mine (game over condition). |
| 2. Interpretation | – The initial state is fully obscured (all cells are '.').
– After a valid action, the game reveals the state of the selected cell.
– If a revealed cell contains a mine ('*'), the game ends immediately — loss.
– If a revealed cell shows a number (e.g., 1, 2), it indicates exactly how many adjacent mines exist (including diagonals), but does not reveal their specific locations.
– A cell is "already revealed" if it has been previously opened and remains unchanged.
– Revealing a cell with value '0' provides no direct mine information but immediately reveals all adjacent unrevealed cells (via propagation). |
| 3. Gameplay | – On each turn, the player selects a cell (row, col) within bounds (0–4).
– If the selected cell is already revealed, the action is invalid and no change occurs.
– If the selected cell is unrevealed (marked '.'), the game reveals its state:
– If it is a mine ('*'), the game ends — loss.
– If it is a safe cell (value '0' or 'n > 0'), the cell is revealed, and the game continues.
– A cell with value '0' triggers automatic revelation of all adjacent unrevealed cells (including those with '.').
– Repeated attempts to reveal already-revealed cells result in invalid actions and no change. |
| 4. Action Effects | – Revealing a cell with value '0' causes all adjacent unrevealed cells to be automatically revealed — this is a key strategic advantage and significantly reduces the search space.
– Revealing a cell with a number 'n > 0' provides partial information: exactly 'n' adjacent mines exist, but their exact positions remain unknown.
– Repeated actions on already-revealed cells are invalid and do not progress the game.
– Invalid actions (e.g., attempting to open a revealed cell) do not cause loss or progress the game. |
| 5. Game Objective | – Win: All unrevealed cells are safely revealed (no mine is exposed), and the final board contains no '.' or '*'.
– Lose: A mine ('*') is revealed during a valid action.
– The game ends immediately upon revealing a mine.
– The game is "solved" when the player has fully revealed all safe cells without exposing any mines. |

Figure 19: An example of a learned environmental rule for Minesweeper environment. All rules are generated by CEL. Blue highlighting marks a few representative cases.

| Rule Composition | Generated-Rule Definition (FrozenLake) |
|---|---|
| 1. Symbol Meanings | – 'P': Player position (starts at a given cell, moves via actions).
– 'H': Hazard (indicates a wall or impassable zone; cannot move into or through).
– 'G': Goal (target destination; game ends when player reaches a goal).
– Empty cell ('. '): Open space; player can move into it. |
| 2. Interpretation | – The player can only move into adjacent cells (up, down, left, right) that are not occupied by 'H' and are within the grid bounds (0–5 in all directions).
– The game state is updated immediately after each valid action.
– 'INFO: You hit a wall!' indicates an invalid move attempting to move into a cell with 'H' or outside the grid boundaries.
– 'INFO: Game solved.' indicates the player has reached a goal ('G') via a valid move.
– 'INFO: Game over.' indicates failure due to either falling into a hazard ('H') or exceeding the maximum number of steps (e.g., 25 turns). |
| 3. Gameplay | – The player takes turns moving in one of four directions: UP, DOWN, LEFT, RIGHT.
– Movement is only allowed to adjacent cells that are empty ('.') or contain a goal ('G').
– The player cannot move into a cell with 'H' or beyond the grid boundaries (row or column must remain within 0–5).
– The game begins at the starting 'P' position and ends when the player reaches a 'G'. |
| 4. Action Effects | – Valid move: Player shifts position to an adjacent cell if the destination is within bounds and not occupied by 'H'.
– Invalid move: Movement into a cell marked 'H' or outside the grid (row or column < 0 or > 5) results in a wall hit and no positional change.
– Reaching a 'G' cell ends the game with a win.
– Falling into a hazard (i.e., moving into a cell marked 'H') results in immediate loss.
– Exceeding the maximum number of steps (e.g., 25 turns) results in a loss. |
| 5. Game Objective | – Objective: Reach a goal ('G') from the starting position ('P') without falling into a hazard or exceeding the maximum number of steps.
– Termination conditions:
– Win: Player reaches a goal ('G') via a valid move.
– Loss: Player falls into a hazard ('H') or exceeds the step limit (e.g., 25 turns). |

Figure 20: An example of a learned environmental rule for FrozenLake environment. All rules are generated by CEL. Blue highlighting marks a few representative cases.

| Rule Composition | Generated-Rule Definition (Sokoban) |
|---|---|
| **1. Symbol Meanings** | – '#': Wall (impassable boundary)
– '.': Empty passage (passable; can be traversed by the player)
– 'P': Player (start position; can move to adjacent empty cells or push a box if applicable)
– 'B': Blue token (initially a movable box; can be pushed if the cell behind it is empty and not occupied by a wall or another box)
– 'G': Green token (goal cell; a box must be moved to a 'G' to win; once a box reaches a 'G', it is marked with '√' and cannot be moved again)
– '√': Final solved state — a box has been successfully moved to a goal cell ('G'). The game is solved when all boxes ('B') are on goal cells ('G') and marked with '√'. |
| **2. Interpretation** | – A valid action (UP, DOWN, LEFT, RIGHT) can only be taken from a cell adjacent to the player ('P').
– If the target cell is empty ('.'), the player moves there.
– If the target cell contains a box ('B'), the box is pushed one cell in the direction of movement **only if** the cell immediately behind the box (in that direction) is empty and not occupied by a wall ('#') or another box.
– The game state is updated after each valid move.
– 'INFO: Your action is valid' → move is allowed and within bounds.
– 'INFO: Your action is invalid' → attempted move is out of bounds, into a wall ('#'), or into a non-empty cell that cannot be entered (e.g., pushing a box into a wall or another box).
– 'INFO: Game solved' → all boxes ('B') have been successfully moved to goal cells ('G') and are marked with '√'; the game ends immediately.
– 'INFO: Game over' → the maximum number of allowed steps (e.g., 15â€"16) has been reached without achieving a solved state; the player loses. |
| **3. Gameplay** | – The player controls a single 'P' that can move one cell at a time in four directions: UP, DOWN, LEFT, RIGHT.
– Movement is allowed only into adjacent empty cells ('.') or to a box ('B') if the box can be pushed (i.e., the cell behind it is empty and not blocked by a wall or another box).
– The player does **not** move tokens directly — instead, movement is used to either reposition the player or push a box.
– Box positions ('B') are initially fixed in space; their final positions are determined by the solved state (all 'B' tokens on 'G' cells).
– The objective is to guide the player's path so that one or more boxes are pushed into goal cells ('G') and marked with '√'.
– Once a box reaches a goal ('G'), it is locked in place and cannot be moved again.
– The player **cannot pull** boxes — only **pushes** are allowed.
– The game ends immediately upon either a win (all boxes on 'G') or a loss (step limit reached or invalid move). |
| **4. Action Effects** | – Valid movement (player or box) updates the game state and advances the turn.
– If a box is successfully pushed into a goal cell ('G'), it is immediately marked with '√' and cannot be moved again.
– If a box is pushed into a wall ('#') or into another box, the move is invalid and no change occurs.
– Invalid movement results in no change and a clear message (e.g., "Cannot move into wall" or "Cannot push box into wall").
– Repeated invalid or looping moves (e.g., cycling between cells without progress) lead to step exhaustion and loss.
– A "game solved" message appears when all boxes are on goal cells ('G') and marked with '√'.
– Upon reaching a cell adjacent to a 'G', the game ends with a win **only if** a box is present on that 'G' — otherwise, the win condition remains tied to box placement. |
| **5. Game Objective** | – **Win Condition**: All boxes ('B') are successfully moved to goal cells ('G') and marked with '√'. This constitutes the final solved state.
– **Loss Conditions**:
– The player exceeds the maximum number of allowed steps (e.g., 15â€"16).
– The player makes an invalid move (e.g., pushing a box into a wall or another box).
– Boxes become stuck in positions where no further movement is possible (i.e., no path exists to any goal).
– The player is unable to make any further valid moves (all boxes are blocked from reaching goals).
– The game ends when either a win is achieved or a loss condition is triggered. |

Figure 21: An example of a learned environmental rule for Sokoban environment. All rules are generated by CEL. Blue highlighting marks a few representative cases.

| Derived Strategic Concept | | Evidence from Model (FrozenLake) |
|---|---|---|
| *Method* | Minimize Steps | Use `efficient, direct paths` to reach the goal. Avoid circular motions or backtracking. |
| | Deadlock Avoidance | If blocked, use vertical or horizontal movement to reposition — e.g., go down to a lower row, then right to reach goal. |
| | Pattern Recognition | Identify pre-existing paths or corridors (e.g., horizontal/vertical lines of '.') to exploit. Use them to move efficiently. |
| *Principle* | Prioritize Pathfinding Over Exploration | Always move toward the nearest goal ('G') using a clear path through open cells ('.'). `Avoid detours or unnecessary exploration`. |
| | Constraint Satisfaction | Avoid Hazards ('H'):** Never move into or adjacent to 'H' cells—they are impassable and lead to immediate loss. |

Figure 22: An example of a learned strategic guideline from the agent's playbook for FrozenLake environment. All rules are generated by CEL. Blue highlighting marks a few representative cases.

| Derived Strategic Concept | | Evidence from Model (Sokoban) |
|---|---|---|
| *Method* | Repositioning | If a box is stuck, check if the player can reposition to push it from another direction — `repositioning is key to unlocking progress`. |
| | Deadlock Avoidance | Avoid creating deadlocks: Never make moves that trap blocks between walls or other blocks. |
| | Greedy Heuristic | Move toward the nearest G or box closest to a G first. |
| | Pattern Recognition | Use symmetry and pattern recognition: In complex layouts, observe recurring patterns (e.g., corner formations) and exploit them to create a "clear path" to goal. |
| *Principle* | Consequential Planning | Plan movement paths: `Always plan a route` to reach a box that can be pushed, ensuring the path behind the box is clear. |
| | Constraint-Based Pruning | Never make a move that results in a box being pushed into a wall or another box — this leads to invalid moves and loss. |
| | Intentional Action | Act only when movement is valid and leads to a new, meaningful state: Every action must contribute to progress. |
| | Local Optimization | Prioritize block alignment with goal: Always aim to position a block (B or P) directly adjacent to the goal (G) before attempting to move it into it. |

Figure 23: An example of a learned strategic guideline from the agent's playbook for Sokoban environment. All rules are generated by CEL. Blue highlighting marks a few representative cases.

