# OpenReview forum: "Cogito, Ergo Ludo: An Agent that Learns to Play by Reasoning and Planning"
_ICLR.cc/2026/Conference — Submitted to ICLR 2026_

### Official Review · Reviewer_gKP8 · 2025-10-26

**Soundness:** 2
**Presentation:** 4
**Contribution:** 3
**Rating:** 4
**Confidence:** 3

**Summary:**

The authors present Cogito, Ergo Ludo (CEL), an LLM-based learning agent that learns to play by using a structured two-pronged algorithm for reasoning and planning, which involves constructing a semi-abstract world model and a “strategic playbook” – essentially policy advice – out of natural language. The authors show that their algorithm outperforms multiple baseline language models that do not use the proposed algorithm.

**Strengths:**

The connection to existing RL literature is thoughtful and sensible, with most relevant literature touched on. The writing style is clear and easy to follow. The idea has perhaps been explored lightly before, but is more thoughtfully integrated into RL than most similar works.

**Weaknesses:**

The improvement of the CEL agent over the Qwen+game rules is modest for some environments. 95% confidence intervals should be plotted, not just average returns. There are unanswered questions about the source of the language descriptions from states.

**Questions:**

Where do the natural language interpretations of game states come from? Are they provided by the environment? Is there a layer between the environment and the LLM where you engineered natural language observations? If so, it would hardly make the agents *tabula rasa.*

In the Generated-Rule Definition tables in the appendix, are the rules in blue the only rules that are learned? Why are they highlighted?

I would guess that the agent performance has high variance, judging by the plots. Why were these experiments not plotted with a proper confidence interval?

**Initial Recommendation:** Reject

**Reasons:** It’s customary to plot confidence intervals for agent performance so we can see the variance of the learning algorithm, please plot them. Also, please address the questions I pose and I will be happy to revise my recommendation to accept.

---

> ### Author Response · Authors · 2025-11-23
> **Rebuttal Reply 1/n (with additional results and paper rewriting)**
>
> Dear reviewer gKP8,
>
> Thank you for your thoughtful feedback and for recognizing our paper’s strengths in terms of (1) clear and easy-to-follow writing, (2) a thoughtful and sensible connection to existing RL literature, and (3) a more thoroughly integrated use of LLM-based reasoning within an RL framework than prior similar works.
>
> We appreciate your constructive suggestions and have addressed your concerns with additional details and analysis.
>
> > **W1: “The improvement of the CEL agent over the Qwen+game rules is modest for some environments.”**
> >
>
> We thank the reviewer for this insightful comment. While the gains may look modest at first glance, they are in fact substantial and informative both quantitatively and conceptually.
>
> **Quantitatively**, CEL achieves sizable absolute improvements over the Qwen+game-rules baseline across all three games. It more than doubles the baseline’s success rate in Minesweeper (54% vs 26%), achieves a strong boost to 84% in the more complex Sokoban puzzle, and reaches near-perfect performance in FrozenLake. In the revised manuscript we also report 95% confidence intervals for all methods, showing that these gains are robust across seeds rather than artifacts of variance.
>
> **Conceptually**, the Qwen+game-rules baseline is a very strong point of comparison: it is an agent equipped with a perfect, hand-coded world model of the game’s dynamics. In contrast, CEL starts with no explicit rules or strategies, must induce both from scratch, and yet ultimately matches or surpasses this rules-provided baseline. We view this as the key result: CEL is not only recovering the environment rules, but also discovering and refining effective strategic play through its playbook, and this learned strategy is strong enough to outperform an agent that is given the rules a priori.
>
> > **Q1: ”Where do the natural language interpretations of game states come from? Are they provided by the environment? Is there a layer between the environment and the LLM where you engineered natural language observations? If so, it would hardly make the agents tabula rasa.”**
> >
>
> We thank the reviewer for raising this important point. To be crystal clear: the “natural language interpretations of game states” are simply the raw textual observations emitted by the environments themselves. There is no intermediate layer where we handcraft or paraphrase observations into higher-level sentences.
>
> As described in Appendix B, we use the TextArena benchmark, where observations are natively given as text-based grids rather than curated natural-language descriptions. For example, the agent receives the raw grid and must learn through trial and error that a symbol like '1' in Minesweeper signifies a specific game mechanic.
>
> This design is fundamental to our claim of learning *tabula rasa*. It forces the agent to autonomously discover not only the game's rules but also the very meaning of the perceptual symbols it observes. We have revised the manuscript to make this observation interface and its role in our setup more explicit.
>
> > **Q2: ”In the Generated-Rule Definition tables in the appendix, are the rules in blue the only rules that are learned? Why are they highlighted? ”**
> >
>
> We thank the reviewer for pointing out this ambiguity. To clarify, the **entire content** of each *Generated-Rule Definition* table is autonomously produced by CEL; the full rulebook is induced from interaction trajectories, not only the highlighted parts.
>
> The blue text is purely a presentational device to draw attention to a few particularly illustrative rules (for example, correctly inferring the meaning of numbered cells in Minesweeper). We agree that this was not clearly stated and could be misleading. In the revised manuscript, we have updated the captions of all relevant figures/tables to explicitly note that all rules are learned and that blue entries are only highlighted examples.

---

> > ### Author Response · Authors · 2025-11-23
> > **Rebuttal Reply 2/n (with additional results and paper rewriting)**
> >
> > > **Q3: ”I would guess that the agent performance has high variance, judging by the plots. Why were these experiments not plotted with a proper confidence interval?
> > Initial Recommendation: Reject Reasons: It’s customary to plot confidence intervals for agent performance so we can see the variance of the learning algorithm, please plot them. Also, please address the questions I pose and I will be happy to revise my recommendation to accept.”**
> > >
> >
> > We thank the reviewer for raising this important point about variance. In the revised manuscript, we have updated **Fig. 11 of the Appendix D** to report **95% confidence intervals** for all methods. At each LLM update step, we evaluate every agent on 32 procedural seeds (with 8 rollouts per seed) and report the mean success rate together with the 95% CI across seeds. In the new plots, the solid purple curve and shaded band denote CEL’s mean and CI, and the dashed green curve and light-green band denote the mean and CI of the Qwen3-4B w/ game rules baseline.
> >
> > The updated figure shows that the gains are robust and practically meaningful. In Minesweeper, CEL reaches a final success rate of 54%, clearly above the baseline band centered around 26%. In Sokoban, CEL attains 84% compared to 73% for the baseline, again with non-overlapping confidence bands at convergence. In FrozenLake, the baseline already performs strongly, so the headroom is smaller, but CEL still consistently climbs to near-perfect performance (97%), and its CI remains above that of the baseline after early training. Thus, once variance is taken into account, the updated curves show that CEL consistently and statistically significantly outperforms the strongest Qwen baseline in all three environments.
> >
> > We sincerely appreciate your constructive suggestions, which helped strengthen our paper a lot. We hope this addresses your concerns and please let us know if you have any questions. Thank you!

---

> > > ### Comment · Reviewer_gKP8 · 2025-11-25
> > >
> > > Thank you, all of my concerns have been addressed. Revising my scores accordingly.

---

### Official Review · Reviewer_WXdM · 2025-10-27

**Soundness:** 3
**Presentation:** 3
**Contribution:** 2
**Rating:** 4
**Confidence:** 3

**Summary:**

This paper introduces Cogito, Ergo Ludo (CEL) , an LLM-centric game-playing agent that learns by reasoning and planning. CEL runs a two-phase loop: during play it performs natural-language lookahead with a Language-based World Model and evaluates options via a Language-based Value Function; after each episode it conducts post-episode reflection to induce/refresh explicit game rules and summarize a strategic playbook, both persisted as text and used in subsequent episodes. Starting tabula rasa (action set only), CEL jointly acts, reasons, and updates its knowledge while also training the LLM parameters (e.g., via GRPO). Evaluated on Minesweeper, Frozen Lake, and Sokoban, CEL discovers rules from interaction, learns effective policies from sparse rewards, and shows generalization beyond seen layouts, pointing to procedure-level transfer.

**Strengths:**

1. The paper is well-structured and easy to follow: the problem setup, agent loop (in-episode reasoning + post-episode reflection), and training details are clearly delineated.

2. Framing gameplay as learning rules and strategies from interaction via an LLM—then persisting them as explicit, reusable knowledge—is a creative twist on world-model RL. The language-based world/value modules and post-episode “playbook” distillation offer a fresh, interpretable alternative to purely latent approaches.

3. The empirical section demonstrates care for robustness: multiple random seeds, ablations on key components (e.g., rule induction, value guidance), and comparisons against sensible baselines.

4. The environments and interfaces appear lightweight and standardized, making it feasible for the community to replicate results and extend the benchmark (new levels or variants) without heavy engineering overhead.

**Weaknesses:**

1. The environments (Minesweeper, FrozenLake, Sokoban) are classic and likely known to pretrained LLMs. This weakens the “tabula rasa” claim: the agent may be recalling prior heuristics instead of actually inducing rules online. The paper should make explicit what prior knowledge the model is allowed to use.

2. The method relies on GRPO to train the LLM, but the optimization setup is under-specified (reward shaping, rollout collection, frequency of distillation from reflection, etc.). It’s difficult to attribute where the gains come from or to reproduce the system.

3. There is no imitation-learning / supervised baseline where the LLM is trained on expert demonstrations. Without this, it’s hard to conclude CEL is the most effective way to make LLMs play these games.

4. The games are small and single-agent puzzles. The paper does not convincingly show (or argue in detail) how CEL scales to harder settings. The contribution would be stronger with either an explicit scaling path or a clearer statement that CEL is currently a controlled proof-of-concept rather than a generally applicable agent paradigm.

**Questions:**

1. What concrete steps ensure the LLM isn’t recalling prior knowledge of Minesweeper/FrozenLake/Sokoban?

2. Could you provide a training diagram + full hyperparameter table and compute budget?

3. How does CEL compare to supervised LLMs trained on a small set of expert demonstrations (BC/DAgger or scripted experts)?

4. To which game classes do you expect CEL to scale, and what modifications are required?

5. Do you expect CEL’s “reason–act–reflect” loop to extend to (a) long-horizon, combinatorial, perfect-information board games like chess and Go, and/or (b) high-bandwidth, partially observed, continuous-control settings like FPS?

6. If CEL cannot straightforwardly handle games like chess/Go or FPS-style environments, how should we interpret its core contribution? Is CEL intended primarily as (i) a self-explaining rule/strategy inducer for small, symbolic environments, or (ii) a generally useful module in a larger agent stack?

7. How would you compose CEL with other methods (tree search, learned motor policies, memory/belief tracking) in a full system?

---

> ### Author Response · Authors · 2025-11-23
> **Rebuttal Reply 1/n (with additional results and paper rewriting)**
>
> Dear reviewer WXdM,
>
> Thank you for your thoughtful feedback and for recognizing our paper’s strengths in terms of (1) a clear and well-structured presentation of the agent loop and training setup, (2) a creative, interpretable twist on world-model RL through language-based rules and playbooks, and (3) a robust empirical evaluation with sensible baselines and lightweight, easily extensible environments.
>
> We appreciate your constructive suggestions and have addressed your concerns with additional experiments and analysis.
>
> > **W1 and Q1: ”What concrete steps ensure the LLM isn’t recalling prior knowledge of Minesweeper/FrozenLake/Sokoban?”**
> >
>
> We thank the reviewer for this critical point. With a pretrained LLM backbone, it is indeed important to be precise about what we mean by *tabula rasa* and what prior knowledge the agent is allowed to use. Our claim is not that the backbone LLM contains no information about classic games, but that, operationally, the agent starts with no **explicit** game rules or strategies (its rulebook and playbook are empty) and all persistent domain knowledge used for decision making must be constructed online through the reason–act–reflect loop.
>
> Concretely, we take the following steps to minimize game-specific recall and to enforce this **explicit-state *tabula rasa*** condition:
>
> **1. Sanitized, name-free prompts.** At the beginning of training, the agent’s context contains only the raw text-based grid observation (e.g., the ASCII/text-based grid description of the current board) and generic instructions about interacting with an unknown environment. We deliberately remove all game names and canonical phrasing (“Minesweeper”, “Frozen Lake”, “Sokoban”, etc.) from the prompts, so there is no direct textual cue that could trigger game-specific heuristics memorized during pretraining. The only information available about the task must be inferred from the observations and the agent’s own evolving rulebook and playbook, which start empty.
>
> **2. Baseline under sanitized prompts.** The “Qwen3-4B w/o game rules” baseline in Figure 3 uses exactly these sanitized, name-free prompts and achieves very low zero-shot success rates across all three environments. This indicates that, under our prompt design and without an explicit rulebook/playbook, the pretrained LLM is not able to simply recall and execute a stored solution.
>
> **3. Ablation: adding the game name.** To further probe how much game-specific prior knowledge can be triggered, we add an ablation “Qwen3-4B w/o game rules + name”, where we keep the prompt identical except for a single sentence stating the environment name (e.g., “You are playing Minesweeper.”). In this condition, zero-shot success rises to 10.94% on Minesweeper, 38.28% on FrozenLake, and 48.05% on Sokoban. In contrast, with our default name-free prompts, the corresponding baseline achieves very low success in all three games (Figure 3). This contrast shows that canonical names do encode strong priors in the pretrained model and supports our choice to use sanitized, name-free prompts when evaluating CEL’s ability to induce rules and strategies from scratch.
>
> > **W2 and Q2: ”Could you provide a training diagram + full hyperparameter table and compute budget?”**
> >
>
> Thank you for asking about the training details. Our experiments are implemented on the open-source rLLM framework for post-training language agents with reinforcement learning. In the revised version, we have added:
>
> **1. Training diagram (Figure 10, Appendix C).** We include a diagram summarizing the full CEL loop: (i) the current policy interacts with the environment to collect trajectories; (ii) the Language-based World Model and Value Function generate rollouts conditioned on the learned rulebook and strategic playbook; (iii) a reflection step updates the rulebook and playbook based on successes and failures; and (iv) a GRPO trainer updates the backbone LLM using the generated reasoning traces, actions, and rewards. This loop is used for all games with the same architecture.
>
> **2. Full hyperparameter table (Table 2, Appendix C).** We provide a table listing, for each environment, the backbone model, learning rate, number of GRPO updates, rollout length, maximum context length, and other key settings. All remaining low-level configurations (e.g., gradient checkpointing, FSDP parameter/optimizer offloading, async rollout engine) follow the default rLLM configuration and are kept fixed across experiments.
>
> **3. Compute budget.** Training a single game configuration (e.g., Minesweeper) requires about 20 wall-clock hours on 8 GPUs (≈160 GPU-hours). The training cost scales approximately linearly with the number of sampled environment steps, and the setup runs on a small cluster of commodity GPUs without any special hardware requirements.

---

> > ### Author Response · Authors · 2025-11-23
> > **Rebuttal Reply 2/n (with additional results and paper rewriting)**
> >
> > > **W3 and Q3: ”How does CEL compare to supervised LLMs trained on a small set of expert demonstrations (BC/DAgger or scripted experts)?”**
> > >
> >
> > We thank the reviewer for highlighting the importance of an imitation-learning / supervised baseline. In the revised manuscript (Appendix E), we add a behavioral cloning (BC) baseline where the same LLM backbone is fine-tuned on 2k expert state–action pairs in the Minesweeper environment generated by a scripted expert (Qwen3-30B). The supervised model fits the training data with high accuracy but achieves essentially zero success on the same test distribution used for CEL, indicating that it largely memorizes the demonstrations and fails to generalize even to new instances of the same game. In contrast, CEL, trained only from sparse success/failure feedback in the same setting, induces explicit rules and a strategic playbook and achieves strong performance on held-out boards.
> >
> > More broadly, our aim in this work is not to claim global optimality over all possible methods, but to show that CEL is an effective and interpretable approach for these sparse-reward grid puzzles, and that it compares favorably to reasonable alternatives such as BC under comparable data and compute.
> >
> > > **Q4: ”To which game classes do you expect CEL to scale, and what modifications are required?”**
> > >
> >
> > We thank the reviewer for this thoughtful question. We expect CEL to scale most naturally to discrete, turn-based games with symbolic or symbolizable state, and to serve as a high-level reasoning module in richer environments. The current work focuses on small single-agent grid puzzles, but the architecture suggests several broader classes:
> >
> > **(i) Larger symbolic puzzles and grid-based MDPs  (e.g., maze and navigation).**
> >
> > Beyond Minesweeper, FrozenLake, and Sokoban, the same CEL architecture applies to maze/navigation domains, 2048-style tile games, or grid-based resource-collection tasks, where states are text/grids and actions are discrete. Our size-scaling and long-horizon experiments already show that CEL can handle larger boards and longer episodes; scaling further mainly requires richer symbolic state encodings (e.g., local patterns/objects) and adjusted horizons, while keeping the reason–act–reflect loop unchanged.
> >
> > **(ii) Combinatorial, perfect-information board games (e.g., Go and chess).**
> >
> > For long-horizon board games, CEL is well-suited as a high-level rule/strategy inducer on symbolic board descriptions. Our preliminary 9×9 Go-like experiment (100-move horizon) shows that CEL can learn non-trivial strategies and outperform Qwen baselines. To move toward richer board games, we would (a) design more structured symbolic encodings of game state (for example, groups, liberties, tactical features) and (b) use a more expressive backbone LLM to support deeper, more precise reasoning. The CEL loop itself remains the same. Scaling up primarily requires a more capable underlying model to cope with the increased combinatorial complexity.
> >
> > **(iii) Multi-agent and stochastic symbolic games (e.g., cooperative grid-worlds and turn-based card games).**
> >
> > For multi-agent, partially stochastic symbolic games (for example, cooperative gridworlds, team-based board games, or arenas with multiple controllable units), CEL’s architecture can be extended in a largely seamless way. CEL’s LWM can be conditioned on agent identities, roles, and shared state, and used to imagine joint or coordinated action sequences, while the LVF evaluates these multi-agent plans. The rulebook/playbook become multi-agent knowledge (conventions, roles, coordination patterns). Handling partial observability or stochastic events mainly requires adding a belief-tracking module that summarizes latent state into text or symbols as extra context.
> >
> > **(iv) High-bandwidth, continuous-control, visually rich environments (e.g., FPS-style games).**
> >
> > For FPS-style or continuous-control games with raw-pixel observations, we do not expect CEL, in its current text-only form, to directly control low-level actions. Instead, we view it as a high-level planner: a perception stack would map observations to compact symbolic descriptions, a learned motor controller would handle fine-grained control, and CEL’s LWM/LVF/rulebook/playbook would operate on the abstract state to induce rules and evaluate strategic goals or tactical plans.
> >
> > Across these game classes, the core algorithmic structure of CEL remains unchanged. Scaling to more complex domains primarily requires (1) richer state representations and surrounding system components (perception, memory, control), and (2) more capable backbone LLMs or multimodal variants so that the language-based world model and value function have sufficient capacity.

---

> > > ### Author Response · Authors · 2025-11-23
> > > **Rebuttal Reply 3/n (with additional results and paper rewriting)**
> > >
> > > > **W4, Q5 and Q6: ”Do you expect CEL’s “reason–act–reflect” loop to extend to (a) long-horizon, combinatorial, perfect-information board games like chess and Go, and/or (b) high-bandwidth, partially observed, continuous-control settings like FPS?
> > > If CEL cannot straightforwardly handle games like chess/Go or FPS-style environments, how should we interpret its core contribution? Is CEL intended primarily as (i) a self-explaining rule/strategy inducer for small, symbolic environments, or (ii) a generally useful module in a larger agent stack?”**
> > > >
> > >
> > > We appreciate the reviewer’s insightful questions and clarifications. We do not claim that the current implementation of CEL is a drop-in, fully general agent for arbitrary games such as full-scale chess, Go, or FPS environments. In this paper, our primary goal is to systematically design and study an LLM-centric agent architecture built around a language-based world model (LWM) and a language-based value function (LVF), coupled with post-episode rulebook/playbook distillation in a reason–act–reflect loop, in controlled symbolic environments with sparse rewards. Within this regime, we show that CEL can induce human-readable rules and strategies from interaction, use them to guide decision making, and generalize beyond the specific layouts seen during training.
> > >
> > > **(a) Long-horizon, combinatorial board games (Go-like).**
> > >
> > > Conceptually, CEL is a natural fit for perfect-information board games as a high-level rule/strategy module: the LWM and LVF can capture human-readable patterns (e.g., local shapes, tactical motifs, endgame principles) and assess long-term consequences, while fine-grained move selection can be delegated to stronger tactical components (larger LLMs, domain-specific engines, or search). To probe this direction empirically, we ran a preliminary experiment on a **9×9 Go-like environment with a horizon of 100 moves**. The agent plays against a rule-aware random opponent that always generates legal moves, and receives a sparse terminal reward based only on the final outcome. In this setting, a Qwen3-4B baseline w/o rules achieves a win rate of about 8%, and a Qwen3-4B baseline with explicit game rules reaches roughly 58%. Starting from a similarly low win rate, CEL improves during training and attains a maximum win rate of 65.6%, consistently outperforming both baselines (**Fig. 9, Section 4.7**). Although this setup uses a simplified opponent and reduced board size, it provides encouraging evidence that the reason–act–reflect loop can learn non-trivial strategies in long-horizon, combinatorial board games.
> > >
> > > **(b) High-bandwidth, partially observed, continuous-control settings (FPS).**
> > >
> > > By contrast, FPS-style environments require rich perception, real-time decision making, and low-level motor control, which are outside the scope of the current work. We do not expect the present text-only implementation of CEL to straightforwardly handle raw-pixel, continuous-control FPS tasks. Instead, we envision CEL’s LWM and LVF as a high-level reasoning layer: a perception and state-estimation stack would convert high-bandwidth observations into compact symbolic descriptions, a learned controller would handle fine-grained motor actions, and CEL would operate on the abstract state to induce rules and to evaluate and propose high-level goals or tactical plans that guide the underlying controller.
> > >
> > > Putting this together, we intend the core contribution of the paper to be the design and empirical study of an LLM-centric agent architecture centered on a LWM and LVF, integrated with explicit rulebook/playbook induction in a reason–act–reflect loop, in controlled symbolic environments with sparse rewards. In this sense, CEL is primarily (i) a self-explaining rule/strategy inducer for small symbolic domains, where we can carefully analyze how an LLM acquires and uses explicit world knowledge. At the same time, we also view CEL as (ii) a generally useful high-level reasoning module in a larger agent stack: its language-level world model and value function are deliberately kept modular so they can be composed with stronger perception, search, and low-level control components in more complex domains.

---

> > > > ### Author Response · Authors · 2025-11-23
> > > > **Rebuttal Reply 4/n (with additional results and paper rewriting)**
> > > >
> > > > > **Q7: “How would you compose CEL with other methods (tree search, learned motor policies, memory/belief tracking) in a full system?”**
> > > > >
> > > >
> > > > Thank you for this important question. In the current paper, CEL already combines reasoning, search, and memory on a small scale: at each decision step, the language-based world model (LWM) performs a one-step lookahead over candidate actions (a shallow tree search in symbolic space), the language-based value function (LVF) ranks these imagined outcomes, and after each episode the reflection phase writes rules and strategies into a persistent rulebook/playbook that acts as long-term, human-readable memory. In a full system, we envision extending these same interfaces to more powerful modules rather than replacing them:
> > > >
> > > > - **Tree search.** For combinatorial planning and board games, classical tree search (e.g., MCTS or other planners) can manage deeper action trees, while CEL provides the “language front-end”: the LWM simulates multi-step counterfactual rollouts in symbolic state space, the LVF scores candidate continuations, and the rulebook/playbook prune rule-violating or clearly poor moves and bias search toward patterns CEL has discovered. The search algorithm handles tree expansion and backups; CEL supplies rich heuristics, priors, and explanations at each node.
> > > > - **Learned motor policies.** In continuous-control or FPS-like environments, we do not expect CEL to output low-level motor commands. A perception and control stack would process raw observations and execute fine-grained actions (e.g., via a learned motor policy), while CEL operates on a compact symbolic description of the scene. The LWM induces rules about cover, line-of-sight, resources, or roles; the LVF evaluates high-level options; and CEL outputs **high-level goals, subgoals, or tactical plans** (e.g., “secure region X”, “retreat to cover”, “coordinate with ally Y”) that the motor policy then implements.
> > > > - **Memory and belief tracking.** In partially observed settings, a memory or belief-tracking module can maintain a latent belief state over the environment and periodically summarize it into text or symbols for CEL. The belief module provides short-term, episode-specific information (“what we currently know”), while CEL’s rulebook/playbook encode longer-term regularities (“how the world tends to work and how we usually play”). The reason–act–reflect loop ties these together: decisions are conditioned on both the current belief and accumulated rules/strategies, and post-episode reflection writes new experience back into the rulebook/playbook.
> > > >
> > > > Overall, we see CEL as a modular high-level reasoning and memory component that sits on top of tree search, learned motor controllers, and belief trackers: its language-based world model, value function, and explicit rulebook/playbook provide structured knowledge, strategic evaluation, and interpretable plans, while specialized modules handle deep search, perception, and low-level control.
> > > >
> > > > Thank you again for your detailed and constructive comments, which guided many of our revisions and new analyses. We believe these changes have made the work much stronger, and we would be glad to answer any remaining questions. Thank you!

---

### Official Review · Reviewer_zbSh · 2025-11-01

**Soundness:** 2
**Presentation:** 3
**Contribution:** 3
**Rating:** 6
**Confidence:** 3

**Summary:**

The paper proposes the CEL agent. A small LLM explicitly summarizes environment rules , then performs one-step lookahead with value estimation to decide actions. On text-based grid environments (Minesweeper, FrozenLake, Sokoban) with sparse terminal rewards and no ground-truth rules provided, the authors report steadily improving success rates and ablations suggesting that continual rule induction is critical. The core value is coupling readable, auditable, language-level knowledge with the action policy, offering an interpretable path to “learning by thinking” on small but challenging tasks.

**Strengths:**

1. Experiments showed that CEL successfully learns to master these tasks
by autonomously discovering their rules and developing effective strategies.
2. Strict evaluation setup (sparse rewards, no ground-truth rules) across three representative grid tasks, with ablations supporting the claim that continual rule induction matters.
3. CEL provide qualitative evidence of the architecture’s unique interpretability, showcasing the comprehensive, human-readable rulebooks and sophisticated strategic heuristics.

**Weaknesses:**

1. Limited evaluation scope: confined to small, text-based grid environments; external validity to larger/complex tasks, higher stochasticity, and distribution shift is unclear.
2. The paper lacks comparisons with baselines like MuZero、Dreamer and other reinforced search discussed in the related work.
3. No verification for induced rules: the loop relies on induction and reuse; there is no explicit verification mechanism (e.g., executable consistency checks, counterexample-driven falsification, precision/recall against ground-truth rules), so systematic errors may persist. I wonder if this will lead to performance bottlenecks in larger systems.

**Questions:**

1. Can you Report results on larger/longer-horizon variants and more stochastic settings.
2.  I wonder if there is a brief failure analysis on hardest cases.
3. Can you add some other baselines like MuZero and Dreamer under matched budgets (steps/compute/tokens) ?
4.  Can you explain why there is no verification for induced rules and whether this will lead to performance bottlenecks in larger systems.

---

> ### Author Response · Authors · 2025-11-23
> **Rebuttal Reply 1/n (with additional results and paper rewriting)**
>
> Dear reviewer zbSh,
>
> Thank you for your thoughtful feedback and for recognizing our paper’s strengths in terms of (1) showing that CEL successfully masters the tasks by autonomously discovering environment rules and effective strategies, (2) using a strict sparse-reward, no-rules evaluation with ablations that support the importance of continual rule induction, and (3) providing qualitatively rich, human-readable rulebooks and sophisticated strategic heuristics that highlight the architecture’s interpretability.
>
> We appreciate your constructive suggestions and have addressed your concerns with additional details and analysis.
>
> > **W1 and Q1: ”Limited evaluation scope: confined to small, text-based grid environments; external validity to larger/complex tasks, higher stochasticity, and distribution shift is unclear.
> Can you Report results on larger/longer-horizon variants and more stochastic settings.”**
> >
>
> We thank the reviewer for raising this important point. We agree that our original benchmarks are relatively small text-based grid environments, and we position CEL as a proof-of-concept for an explicit language-based world-model with playbook architecture. In the revised manuscript, we add experiments that probe larger / longer-horizon variants, stronger distribution shift, and more complex, stochastic settings. A concise summary of these additional results is given in Table below.
>
> **(1) Larger / longer-horizon variants.**
> Beyond the original 5×5 Minesweeper setting, we train CEL only on 5×5 boards and then evaluate the trained agent **zero-shot on larger boards** (6×6, 7×7, 8×8). In this more challenging regime, CEL achieves 41.4% success over 256 instances, compared to 53.5% on in-distribution 5×5 boards and 50.4% on 256 unseen 5×5 layouts. As expected, the task becomes harder as the board grows, but performance does not collapse, indicating that the induced rulebook and playbook generalize beyond the training size.
> We also scale FrozenLake to a 10×10 grid with a **50-step horizon**; in this larger, longer-horizon variant CEL attains 77.3% success, clearly outperforming a Qwen3-4B w/ game-rules baseline (65.2%). In addition, we study a 9×9 Go-like environment with a **100-move horizon** (see below), further probing CEL in much longer, combinatorial board-game settings.
>
> **(2) Distribution shift.**
>
> Our evaluation already considers both intra-game and inter-game distribution shift. For **intra-game shift**, we evaluate CEL on 32 seeds with layouts entirely unseen during training (256 instances total); the success curves on these held-out layouts remain close to in-distribution performance, suggesting that CEL has learned transferable game principles rather than memorizing specific levels.
>
> For a stronger form of shift, Figure 8 presents an **inter-game generalization** study: a Minesweeper-trained agent adapting to FrozenLake and a FrozenLake-trained agent adapting to Minesweeper with **frozen model weights**, relying only on iterative refinement of the rulebook and strategic playbook. Both settings show robustness to substantial changes in layout and even in underlying game dynamics.
>
> **(3) More complex and stochastic settings.**
>
> To further probe CEL in a more complex, long-horizon, and stochastic/adversarial setting, we run a preliminary experiment on a **9×9 Go-like environment with a 100-move horizon**. The agent plays against a rule-aware random-play opponent that always generates legal moves according to the Go rules but samples uniformly from its legal actions, and receives a sparse terminal reward based solely on the final game outcome.
> In this setting, a Qwen3-4B baseline without rules achieves a win rate of about 8%, and a Qwen3-4B baseline with explicit rules reaches roughly 58%. Starting from a similarly low win rate, CEL improves during training and attains a maximum win rate of 65.6%, consistently outperforming both baselines (Figure 9 in Section 4.7). Although this setup uses a simplified opponent and reduced board size, it provides encouraging evidence that CEL’s reason–act–reflect loop can learn non-trivial strategies in long-horizon, combinatorial board games, and supports viewing CEL as a promising high-level reasoning module for more complex and stochastic domains.
>
> | Aspect | Setting | CEL performance | Baseline |
> | --- | --- | --- | --- |
> | **Larger / longer-horizon variants** | 5×5 → 6x6 - 8×8 Minesweeper (zero-shot scaling) | 41.4% | 53.5% (same CEL on 5×5 train layouts) |
> |  | FrozenLake 10×10 (50-step horizon) | 77.3% | 65.2% (Qwen3-4B w/ game rules) |
> | **Distribution shift** | Intra-game shift (held-out 5×5 layouts on Minesweeper) | 50.4% | 53.5% (same CEL on 5×5 train layouts) |
> |  | Inter-game shift (Trained on Minesweeper, test on FrozenLake) | 97.3% | N/A |
> |  | Inter-game shift (Trained on FrozenLake, test on Minesweeper) | 46.9% | N/A |
> | **Stochastic / adversarial setting** | Go-like 9×9 (100 moves) | 65.6% | 57.8% (Qwen3-4B w/ game rules) |

---

> > ### Author Response · Authors · 2025-11-23
> > **Rebuttal Reply 2/n (with additional results and paper rewriting)**
> >
> > > **Q2: ”I wonder if there is a brief failure analysis on hardest cases.”**
> > >
> >
> > We thank the reviewer for this insightful question. By manually inspecting failed episodes at the end of training, we most often observe two types of errors.
> >
> > First, the agent can over-generalize from a small number of sparse experiences and induce rules that are too coarse. For example, in FrozenLake we sometimes see playbook entries that effectively discourage visiting an entire region of the grid after a few unlucky failures there, even though some layouts require crossing that region to reach the goal. In such cases the induced strategy is overly conservative: it avoids the only viable path and the agent gets stuck in safe but unproductive parts of the map.
> >
> > Second, some of the hardest failures arise in layouts that contain long, irreversible action chains. A move that clearly improves the local situation (such as pushing a box closer to a goal tile) can still commit the agent to a configuration that only becomes a deadlock many steps later and is difficult to repair. We find that the relevant local rules (how boxes move, when a position is blocked, etc.) are already present, but the decision procedure occasionally prefers locally attractive actions that subtly reduce the number of viable solution branches and eventually lead to an unsolvable state.
> >
> > Taken together, these patterns suggest two main limitations of the current instantiation of CEL: the early induced playbook can be sensitive to rare but misleading experiences, and action selection can struggle with rare long-horizon traps in environments with irreversible dynamics. We view both as opportunities for future work, for example by using stronger backbone LLMs and strengthening long-horizon reasoning through deeper or more selective lookahead in domains where long-term consequences are critical.
> >
> > > **W2 and Q3: ”Can you add some other baselines like MuZero and Dreamer under matched budgets (steps/compute/tokens) ?”**
> > >
> >
> > We thank the reviewer for this thoughtful suggestion. MuZero and Dreamer are indeed canonical learned world-model agents, but they target a rather different regime: compact latent dynamics models optimized for predictive accuracy and control performance, typically under very large-scale interaction. In contrast, CEL is explicitly designed around an **interpretable, language-based** world model and playbook that produce natural-language rules and value judgements. Our primary goal in this work is to study whether such “value in words” and explicit rules can be learned and used effectively in symbolic text environments, rather than to compete head-to-head with MuZero/Dreamer on raw control metrics.
> >
> > Implementing a competitive MuZero/Dreamer-style agent in our text-only setting in a way we would regard as fair and informative would also be a substantial engineering effort. It would require additional networks to encode free-form text states into compact latents, plus separate dynamics, reward, and value heads trained from scratch on relatively small, sparse-reward domains. Moreover, “**matching budgets**” across these paradigms is inherently asymmetric: MuZero/Dreamer are designed to amortize small networks over very high-throughput environment interaction, whereas CEL spends most compute on large LLM forward/backward passes and generates comparatively few interaction steps. Defining a single notion of “equal steps/compute/tokens” that is fair to both is therefore non-trivial and, in our view, beyond the scope of this initial study.
> >
> > In this paper, we therefore control for the backbone model and training pipeline and compare CEL against strong **LLM-centric** baselines under the same text-only interface. Our aim is not to position CEL as a drop-in replacement for MuZero or Dreamer, but to demonstrate that an explicit language-based world model and strategic playbook can be learned from sparse interaction in symbolic text games and provide useful, interpretable guidance.

---

> > > ### Author Response · Authors · 2025-11-23
> > > **Rebuttal Reply 3/n (with additional results and paper rewriting)**
> > >
> > > > **W3 and Q4: ”Can you explain why there is no verification for induced rules and whether this will lead to performance bottlenecks in larger systems.”**
> > > >
> > >
> > > We thank the reviewer for this excellent question. Our choice not to use an explicit ground-truth verifier is deliberate and rooted in the *tabula rasa* setting: we do not assume access to true rules that could be used for external checking. Instead, CEL implements a **conservative rulebook merge** as an implicit verification mechanism.
> > >
> > > Concretely, we found that summarizing rules from a single trajectory is risky, as it can be dominated by anomalous episodes. After each episode, newly proposed rules are therefore **not** **directly adopted**. Rather, an additional LLM call with a dedicated merge prompt compares them against the existing rulebook and only integrates clear, non-contradictory refinements, preferring stability when there is doubt. This review step mitigates catastrophic drift and encourages the rulebook to evolve cautiously over time. The merge prompt used in our experiments is provided in Figure 17 of Appendix L.
> > >
> > > For larger systems, our consistency verification also helps maintain the relative stability of the rules without requiring ground-truth access. We also see two natural scaling paths: (i) replacing the current simple merge prompt with a more sophisticated induction / consolidation procedure, and (ii) leveraging more capable backbone LLMs, which can improve both the quality of induced rules and the reliability of the merge step. Together, these provide a way for CEL’s explicit rulebase to grow in complexity without becoming a performance bottleneck.
> > >
> > > We greatly appreciate your feedback and believe that these new experiments and clarifications strengthen our paper. We hope this addresses your concerns and please let us know if you have any questions. Thank you!

---

### Official Review · Reviewer_PTDm · 2025-11-03

**Soundness:** 4
**Presentation:** 3
**Contribution:** 4
**Rating:** 8
**Confidence:** 3

**Summary:**

This paper presents a novel approach to leverage LLMs to learn world models directly in natural language.
The proposed agent architecture uses a loop of planning and interaction, and world model updates through rule induction from episodic experience.
The authors evaluate their method directly using text-based versions of Minesweeper, Sokoban and Frozen Lake.

**Strengths:**

- The paper proposes quite an interesting approach to world modeling and planning. The paper is generally well written.
- The paper proposes a novel way to do model-based learning in language space and leverage LLMs/GRPO to learn interactively.
- The natural interpretable process provides great insight about the abilities the agent learned while building a world model. Authors showed generalization capabilities, and strategy and rule induction directly from interaction and sparse rewards.
- The authors show quantitatively that the agent performance improves significantly over zero-shot approaches and include ablations that show that the reflection/thinking part of their framework in contributing importantly to the performance gain.

**Weaknesses:**

- In terms of quantitive results, I’m only concerned about the lack of a fine-tuning baseline. Wouldn’t the naive thing be to directly use the reward and GRPO to learn to the play the game? I would expect this to be part of the baselines.
- The evaluation domains are limited to simple games and I’d be interested to understand how this would scale to more complicated domains. However, the proposed idea, imho, is interesting and novel enough that these domains show potential.
- However, it would be really interesting to understand how the performance behaves when the game size grows. Typical RL would struggle. I wonder if the generalization capabilities would shine is this case. Is this considered within the 256 instances used for evaluating generalization or just the change of layout?

**Questions:**

It's very interesting to me the use of natural language to describe a kind of qualitative value function. I wonder if this approach can give us an expressive enough value for more complicated settings? Wouldn't the model be limited by the discrete nature of language?

---

> ### Author Response · Authors · 2025-11-23
> **Rebuttal Reply 1/n (with additional results and paper rewriting)**
>
> Dear reviewer PTDm,
>
> Thank you for your thoughtful feedback and for recognizing our paper’s strengths in terms of (1) an interesting and well-written approach to world modeling and planning, (2) a novel way to do model-based learning in language space by leveraging LLMs/GRPO interactively, and (3) an interpretable learning process that induces rules and strategies from sparse interaction while significantly improving over zero-shot approaches.
>
> We appreciate your constructive suggestions and have addressed your concerns with additional details and analysis.
>
> > **W1: ”In terms of quantitive results, I’m only concerned about the lack of a fine-tuning baseline. Wouldn’t the naive thing be to directly use the reward and GRPO to learn to the play the game? I would expect this to be part of the baselines.”**
> >
>
> We appreciate the reviewer for pointing out this natural fine-tuning baseline. We agree that a natural baseline is to directly fine-tune the LLM with GRPO on the environment reward, without any explicit reasoning or rule memory. We have in fact implemented exactly this “action-only” agent, which outputs only actions and is trained with the same GRPO objective as CEL (see Figure 12 in Appendix F).
>
> In our experiments, this baseline consistently failed to learn. As analyzed in Appendix F, rollout outcomes for the action-only agent are extremely polarized: for a given seed and batch size (8–64 rollouts), either all rollouts in the batch succeed or all fail. Consequently, the number of “partially successful” rollouts within a batch is essentially zero. Since GRPO (like other preference-based RL algorithms) relies on within-prompt differences in return to construct a learning signal, this degenerate outcome distribution starves the algorithm of gradient signal and training collapses.
>
> This failure mode is precisely what CEL is designed to address: by inducing explicit game rules and strategic playbook entries and optimizing over full reasoning traces (not just actions), CEL produces a much richer and more diverse distribution of rollouts. Different reasoning chains for the same state lead to different returns, providing the mixed-success batches that GRPO needs in order to make progress.
>
> > **W2 and W3: ”The evaluation domains are limited to simple games and I’d be interested to understand how this would scale to more complicated domains. However, the proposed idea, imho, is interesting and novel enough that these domains show potential.
> However, it would be really interesting to understand how the performance behaves when the game size grows. Typical RL would struggle. I wonder if the generalization capabilities would shine is this case. Is this considered within the 256 instances used for evaluating generalization or just the change of layout?”**
> >
>
> We appreciate the reviewer’s interest in how CEL scales beyond simple grid games and how its generalization behaves as game size grows. We agree that our current domains are relatively simple, and we intend this work as a proof-of-concept for an explicit language-based world model + playbook architecture. In the main Minesweeper setting, CEL reaches 53.5% success on in-distribution 5×5 boards and 50.4% on 256 unseen 5×5 layouts, indicating strong layout-level generalization at a fixed board size and a substantial improvement over the rules-provided baseline.
>
> Following your suggestion, we additionally ran a preliminary **size-scaling** experiment. We train CEL only on 5×5 boards and then evaluate the trained agent **zero-shot on 256 larger boards** (6×6, 7×7, 8×8). In this more challenging setting, CEL still achieves 41.4% success without any extra training. Although the task naturally becomes harder as the board grows, the agent maintains a strong level of performance, which we see as encouraging evidence that the learned rulebook and playbook do generalize beyond the training size.
>
> Conceptually, CEL’s induced rules are expressed in a local, size-agnostic form (e.g., relating numbers to neighboring tiles rather than to a particular board geometry), and the overall architecture is modular: the language-based world model and playbook can be paired with richer perception modules or low-level controllers. This makes CEL a natural candidate for scaling to larger and more visually complex domains, where it can serve as an interpretable high-level reasoning and planning layer on top of more powerful environment-specific components. In the revised manuscript, Appendix J (“**Extending CEL to Broader Classes of Games**”) provides a more detailed discussion of how the architecture can be composed with additional modules to handle richer, higher-dimensional environments.

---

> > ### Author Response · Authors · 2025-11-23
> > **Rebuttal Reply 2/n (with additional results and paper rewriting)**
> >
> > > **Q1: ”It's very interesting to me the use of natural language to describe a kind of qualitative value function. I wonder if this approach can give us an expressive enough value for more complicated settings? Wouldn't the model be limited by the discrete nature of language?”**
> > >
> >
> > We thank the reviewer for raising this conceptual question. In CEL, natural language does **not** replace the environment’s scalar value function: the environment still provides numeric rewards (success/failure), and GRPO is applied to full trajectories using these returns. The language-valued feedback is an additional, interpretable signal attached to states, rules, and reasoning traces: the model not only assigns a label such as “high/medium/low value”, but also explains *why* in natural language. Internally, these descriptions are encoded into **high-dimensional continuous embeddings rather than three discrete symbols**, so the information they convey is much richer than a small finite label set.
> >
> > Empirically, we find that the induced language-valued feedback is quite expressive in our domains. As illustrated in Figure 13 of Appendix H, the model’s value descriptions spontaneously refer to multiple qualitative dimensions: goal proximity and reachability, safety versus risk, progress versus dead ends and irreversibility, existence of multi-step winning plans, uncertainty about goal reachability, positional control of key regions, and timing/opportunity windows for high-value targets. In other words, this “**value in words**” captures fine-grained structure that would be hard to encode with a small discrete value vocabulary, and it complements the scalar reward by shaping the induced rules and strategic playbook.
> >
> > We agree that in very high-bandwidth, continuous-control settings, a purely linguistic value signal would likely not be sufficient as the only critic. Our intended use in such settings is to keep language-valued feedback at the abstract planning level (constraints, preferences, strategic trade-offs), while using continuous critics or policies for low-level control. In the revised manuscript, **Appendix H (“Value in Words: Language-Based Feedback from CEL”)** provides concrete examples of LVF outputs and discusses how this language-valued component is expressive enough for the discrete, symbolic puzzles we study, while naturally serving as a high-level guidance signal in more complex environments.
> >
> > We are especially grateful for your positive evaluation and for recognizing our work as a novel and interesting approach to world modeling and planning in language space. We hope this addresses your concerns and please let us know if you have any questions. Thank you again for your thoughtful review and strong support of the paper.

---

### Author Response · Authors · 2025-11-23
**General Response 1/n**

Dear Reviewers and ACs,

Thank you again for overseeing our submission and for your thoughtful feedback. In this response, we summarize the main concerns raised and the corresponding clarifications and new experiments in the revised version. We first thank the reviewers for recognizing our work as “a novel way to do model-based learning in language space” that provides “quite an interesting approach to world modeling and planning” (PTDm), “a creative twist on world-model RL” and “a fresh, interpretable alternative to purely latent approaches” (WXdM), and as an LLM-based agent whose “connection to existing RL literature is thoughtful and sensible” and whose “writing style is clear and easy to follow” (gKP8). We also appreciate that CEL was recognized as an agent that “successfully learns to master these tasks by autonomously discovering their rules and developing effective strategies” under a “strict evaluation setup (sparse rewards, no ground-truth rules)” and provides “comprehensive, human-readable rulebooks and sophisticated strategic heuristics” (zbSh, PTDm).

To address the concerns raised, we have expanded and clarified our work by: (1) formalizing the problem setup and explicit-state *tabula rasa* condition and clarifying the source of observations, (2) strengthening the empirical evaluation and baselines, including larger and more challenging settings, (3) providing detailed training and rule-maintenance procedures, and (4) expanding the discussion of scope, limitations, and connections to RL. Below, we detail our responses:

**1: Clarified problem setup, *tabula rasa* condition, and observations** *(reviewers gKP8, WXdM)*

- **Explicit-state *tabula rasa* (Section 3.2).** We now explicitly define an explicit-state *tabula rasa* setting: despite the pretrained backbone, the agent’s explicit game knowledge starts empty (rulebook and playbook are empty) and all persistent game knowledge is constructed online through the reason–act–reflect loop.
- **Observation interface (Appendix B).** We clarify that CEL observes only the raw text-based grid outputs of the environments, with no additional handcrafted natural-language wrapper.
- **Controlling prior knowledge (Appendix G).** To limit recall of game-specific priors, we use sanitized, name-free prompts that do not mention the game name. We add an ablation baseline (“Qwen3-4B w/o rules + name”) which only adds the game name; naming alone substantially boosts zero-shot performance, confirming that our default name-free setup forces CEL to induce its own rules and strategies.

**2: Strengthened empirical evaluation, baselines, and scaling experiments** *(reviewers WXdM, zbSh, PTDm)*

- **Statistical robustness (Appendix D).** We now report 95% confidence intervals and clearly describe our evaluation protocol (32 seeds × 8 rollouts) for all methods, demonstrating that CEL’s gains over Qwen w/ game rules are statistically robust.
- **Imitation-learning baseline (Appendix E).** We add a behavioral cloning (BC) baseline using the same backbone fine-tuned on expert state–action pairs. The BC model fits demonstrations but fails to generalize to unseen layouts, in contrast to CEL’s strong transfer via explicit rules and playbooks.
- **Scalability and distribution shift (Section 4.6).** We include larger and longer-horizon experiments (training on 5×5 Minesweeper and testing zero-shot on 6×6–8×8 boards, and a 10×10 FrozenLake) and study both intra-game and cross-game distribution shift (e.g., training on Minesweeper and adapting to FrozenLake) with frozen model weights and updated rulebooks/playbooks.
- **More complex game (Section 4.7).** We introduce a 9×9 Go-like two-player environment with a 100-move horizon, where CEL’s win rate rises to 65.6%, outperforming rule-free and rule-aware Qwen baselines while retaining interpretable rules and value explanations.

**3: Added training diagram, hyperparameters, compute, and rule verification** *(reviewers WXdM, zbSh)*

- **Training diagram (Appendix C.1).** We add a training diagram summarizing the full CEL loop: environment interaction, language-based world/value rollouts, post-episode reflection to update the rulebook/playbook, and GRPO updates to the backbone LLM.
- **Hyperparameters and compute budget (Appendix C.2).** We provide a complete hyperparameter table and report our compute budget, with other settings following the rLLM defaults for reproducibility.
- **Conservative rulebook merge (Appendix L).** We detail a conservative rulebook-merge procedure as an implicit rule-verification mechanism: newly induced rules are merged with existing ones via a dedicated merge prompt that prefers stability. The exact merge prompt is included in the appendix.

---

> ### Author Response · Authors · 2025-11-23
> **General Response 2/n**
>
> **4: Expanded discussion of scope, limitations, and RL context** *(reviewers WXdM, zbSh, PTDm)*
>
> - **Language-valued feedback (Appendix H).** We provide additional qualitative examples and discussion of language-valued feedback from the LVF, showing how textual value judgments capture nuanced notions such as risk, safety, progress, and dead ends that complement scalar rewards.
> - **Scope and role of CEL (Appendix J).** We expand the discussion and a dedicated appendix to position CEL as a proof-of-concept for an explicit language-based world-model and playbook architecture in small symbolic environments, and as a high-level reasoning module that can be composed with tree search, low-level control, and belief tracking in more complex systems.
> - **Failure analysis and limitations (Appendix I).** We add a brief failure analysis that summarizes typical error modes in the hardest settings (e.g., over-conservative rules in FrozenLake and long-horizon traps in Sokoban), clarifying the main challenges of the current instantiation of CEL and suggesting directions such as stronger backbones and deeper or more selective lookahead for future work.
> - **Relation to latent world-model RL (Section 2).** We refine our comparison to MuZero/Dreamer-style latent world-model agents, noting that they operate in a different regime (compact latent dynamics under massive interaction), whereas CEL targets symbolic text environments and emphasizes transparency and reusable, human-readable knowledge. Consequently, our empirical comparisons focus on strong LLM-centric baselines under the same text-only interface, and we view MuZero-style approaches as complementary rather than direct competitors.
>
> **Additional revisions**
>
> - Improved figure captions and appendix organization.
> - Minor edits throughout to clarify notation, define the game classes more precisely, and state the claims and limitations of CEL more explicitly.
>
> We sincerely appreciate your constructive suggestions, which have helped us substantially strengthen the paper. We are happy to provide any further clarifications. Thank you for your time and consideration.

---

### Author Response · Authors · 2025-12-01
**Author summary of post-discussion updates and clarifications**

Dear AC and reviewers,

In light of the recent policy that reverts reviews to their pre-discussion state, we would like to provide a short author summary that contextualizes the current reviews in terms of the clarifications and new experiments added during the discussion phase.

Our paper introduces **Cogito, Ergo Ludo (CEL)**, an LLM-centric agent that learns to play by reasoning and planning: starting from an explicit-state tabula rasa condition (empty rulebook and playbook), CEL iteratively induces language-level rules and a strategic playbook from raw textual interaction, and uses them for in-episode reasoning and value-guided planning. The reviews highlight CEL as **a novel, interpretable approach to world modeling and planning in language space**, and as an agent that successfully masters sparse-reward text games by **autonomously discovering rules and strategies**.

All technical clarifications and new experiments on the tabula rasa setup and priors, baselines and statistical robustness, scaling to harder settings, training details, and the scope/limitations and relation to RL are documented in our earlier responses and in the revised draft, with explicit pointers to the relevant sections and appendices. Below we briefly summarize how the reviewers’ impressions changed after these updates and the discussion.

**Reviewers’ impressions after discussion**

- **Reviewer PTDm (rating 8)** initially raised concerns about the lack of a fine-tuning baseline and about scaling beyond simple games, but was otherwise strongly positive about the novelty and clarity of CEL. The added GRPO action-only baseline and the new scaling experiments address these points directly.
- **Reviewer gKP8 (rating 4 → 8)** requested stronger empirical reporting and clearer presentation of the observation setting and figures. After we added 95% confidence intervals, clarified the observation pipeline and sanitized prompts, and revised the captions, gKP8 wrote in their follow-up comment that *“Thank you, all of my concerns have been addressed. Revising my scores accordingly.”* and explicitly stated that they were revising their overall score from 4 to 8. While the numerical score currently shown corresponds to the pre-discussion phase, this comment records their updated, post-discussion assessment.
- **Reviewer zbSh (rating 6)** highlighted CEL’s strict sparse-reward, no-rules setup and interpretability, while asking for larger/longer-horizon and more stochastic settings, distribution-shift experiments, failure analysis, and discussion of conventional RL and rule verification. We extended the paper with size scaling, a larger FrozenLake, cross-game transfer, a Go-like 9×9 environment with 100 steps, an added failure analysis, and a conservative rulebook merge procedure to address these points.
- **Reviewer WXdM (rating 4)** found the paper well-structured and easy to follow, and highlighted CEL’s creative framing of gameplay as inducing rules and strategies via an LLM, as well as the robustness of our empirical evaluation. They raised questions about the precise tabula rasa definition, prior knowledge control, training details, supervised baselines, scaling, and the role of CEL relative to broader RL. **We added clarifications and additional experiments to address these concerns**, including the explicit-state tabula rasa formalization, sanitized prompts and name ablations, the added training diagram and hyperparameters, the behavioral cloning baseline, additional scaling experiments, and the clarified scope and positioning in Section 2 and the appendices.

In summary, the current version of the paper directly responds to the key technical and experimental concerns raised in the initial reviews. One reviewer, who had originally given a score of 4, explicitly raised their overall score to 8 after stating that all of their concerns had been addressed by the revisions and additional results. We respectfully believe that, in its current form, the paper offers a novel and carefully executed contribution on explicit language-based world models and planning.

We are sincerely grateful to the AC and all reviewers for the time and effort they have devoted to this work, and for the thoughtful feedback that has helped us substantially improve the paper.

---

### Meta-Review · Area_Chair_wxTg · 2026-01-05

**Summary:**

This paper propose CEL as an agent learns to play by reasoning and planning. CEL is a new type of intelligent agent that continuously refines its understanding of the environment and decision-making abilities by predicting action → language reasoning → reflection to generate explicit rules and strategies, thereby demonstrating high sample efficiency and interpretable learning capabilities in environments with unknown rules and sparse rewards.
Basically, the idea lacks novelty. The only novel part is the world model paradim in the problem solving. Other parts, such as " language reasoning → reflection to generate explicit rules and strategies" are all well developed in the literature, such as the following one that summarizes language strategies for in-context learning and searching in the self-play:
Strategist: Self-improvement of LLM Decision Making via Bi-Level Tree Search, ICLR'25.
In the review, three reviewers have major concerns on datasets used are very simple ones:
WXdM: "The games are small and single-agent puzzles. The paper does not convincingly show (or argue in detail) how CEL scales to harder settings. The contribution would be stronger with either an explicit scaling path or a clearer statement that CEL is currently a controlled proof-of-concept rather than a generally applicable agent paradigm."
zbSh: "Limited evaluation scope: confined to small, text-based grid environments; external validity to larger/complex tasks, higher stochasticity, and distribution shift is unclear."
PTDm: "However, it would be really interesting to understand how the performance behaves when the game size grows. Typical RL would struggle. I wonder if the generalization capabilities would shine is this case. Is this considered within the 256 instances used for evaluating generalization or just the change of layout?"
 However, the authors do not provide real complex setting validation in the rebuttal: "we position CEL as a proof-of-concept for an explicit language-based world-model with playbook architecture."
I don't think this can solve reviewers' concern.

**Reviewer Concerns:**

Three reviewers have major concerns on datasets used are very simple ones:
WXdM: "The games are small and single-agent puzzles. The paper does not convincingly show (or argue in detail) how CEL scales to harder settings. The contribution would be stronger with either an explicit scaling path or a clearer statement that CEL is currently a controlled proof-of-concept rather than a generally applicable agent paradigm."
zbSh: "Limited evaluation scope: confined to small, text-based grid environments; external validity to larger/complex tasks, higher stochasticity, and distribution shift is unclear."
PTDm: "However, it would be really interesting to understand how the performance behaves when the game size grows. Typical RL would struggle. I wonder if the generalization capabilities would shine is this case. Is this considered within the 256 instances used for evaluating generalization or just the change of layout?"
 However, the authors do not provide real complex setting validation in the rebuttal: "we position CEL as a proof-of-concept for an explicit language-based world-model with playbook architecture."
I don't think this can solve reviewers' concern.

**Reviewer Scores:**

Two reviewers are positve, two are negative. It seems one reviewer is planning to raise score after rebuttal. I do not think the remaining reviewer would be convinced after rebuttal.

---

### Decision · Program_Chairs · 2026-01-26

Reject